

# Investigation of the mixing layer height derived from ceilometer measurements in the Kathmandu Valley and implications for local air quality

Andrea Mues[1], Maheswar Rupakheti[1], Christoph Münkel[2], Axel Lauer[3], Heiko Bozem[4], Peter Hoor[4], Tim Butler[1], and Mark Lawrence[1]

[1]Institute for Advanced Sustainability Studies (IASS), Potsdam, 14467, Germany
[2]Vaisala GmbH, Hamburg, 22607, Germany
[3]Deutsches Zentrum für Luft- und Raumfahrt (DLR), Institut für Physik der Atmosphäre, Oberpfaffenhofen, Germany
[4]Institute for Atmospheric Physics, Johannes Gutenberg University Mainz, Mainz, Germany

*Correspondence to:* A. Mues (andrea.mues@iass-potsdam.de)

**Abstract.** In this study one year of ceilometer measurements taken in the Kathmandu Valley, Nepal, in the framework of the SusKat project (A Sustainable Atmosphere for the Kathmandu Valley) were analyze to investigate the diurnal variation of the mixing layer height and its dependency on the meteorological conditions. In addition, the impact of the mixing layer height on the temporal variation and the magnitude of the measured black carbon concentrations are analysed for each season. Based

on the assumption that black carbon aerosols are vertically well mixed within the mixing layer and the finding that the mixing layer varies only little during night time and morning hours, black carbon emission fluxes are estimated for these hours and per month. Even though this method is relatively simple, it can give an observationally based first estimate of the black carbon emissions in this region, especially illuminating the seasonal cycle of the emission fluxes.

In all seasons the diurnal cycle of the mixing layer height is typically characterized by low heights during the night and
maximum values during in the afternoon. Seasonal differences are found in the absolute mixing layer height values and the duration of the typical daytime maximum. During the monsoon season a diurnal cycle has been observed with the smallest amplitude, with the lowest daytime mixing height of all seasons, and also the highest nighttime and early morning mixing height of all seasons. These characteristics can mainly be explained with the frequently present clouds and the associated reduction in incoming solar radiation and outgoing longwave radiation.

In general, the black carbon concentrations show a clear anticorrelation with mixing layer height measurements, although this relation is less pronounced in the monsoon season. The daily evolution of the black carbon diurnal cycle differs between the seasons, partly due to the different meteorological conditions including the mixing layer height. Other important reasons are the different main emission sources and their diurnal variations in the individual seasons. The estimation of the black carbon emission flux for the morning hours show a clear seasonal cycle with maximum values in December to April. Compared to
the emission flux values provided by different emission databases for this region, the here estimated values are considerably higher. Several possible sources of uncertainty are considered, and even the absolute lower bound of the emissions based on





our methodology is higher than in most emissions datasets, providing strong evidence that the black carbon emissions for this region have likely been underestimated in modelling studies thus far.

# 1 Introduction

The project "A Sustainable Atmosphere for the Kathmandu Valley" (SusKat) (Rupakheti et al., 2016) initiated by the Institute
for Advanced Sustainability Studies (IASS) and the International Center for Integrated Mountain Development (ICIMOD) aims at a better understanding of the observed severe air pollution in the Kathmandu Valley and its surroundings. Adverse air quality is a major environmental and health concern in Nepal, especially in the urban areas of the Kathmandu Valley (Aryal et al., 2009; Panday and Prinn, 2009; Sharma et al., 2012). The rapidly expanding urban areas and the constantly increasing traffic volume in the Kathmandu Valley have resulted in a visible degradation of the air quality over the last decades. As part
of the SusKat project the atmospheric characterization campaign SusKat-ABC was conducted in Nepal from December 2012 through June 2013, during which a large number of chemical compounds and meteorological parameters were measured at several sites in the Kathmandu Valley and other parts of Nepal. The first measurement results highlight the severe air pollution and the need for a better understanding of the emissions and the meteorological and chemical processes resulting in such high pollution levels in the valley (e.g. Putero et al., 2015; Sarkar et al., 2016; Chen et al., 2016).

For local air quality the so-called planetary boundary layer (PBL) is of key importance. This layer is directly affected by the underlying surface and responds to surface forcing (e.g. friction, evaporation, heat transport and emission of pollutants) with a timescale of about an hour or less (Stull, 1988). The thickness of the planetary boundary layer varies in time and space, typically ranging from about one hundred meters to a few kilometres. The structure of the planetary boundary layer has important implications for air quality as most air pollutants are emitted into this layer, get transported horizontally and
vertically and affect the environment there. For these reasons it is particularly valuable to have information on the vertical structure and characteristics of the planetary boundary layer. A general characteristic of this layer is the distinct diurnal cycle of the temperature, which is less distinct for the free atmosphere. The incoming solar radiation is absorbed at the ground which reacts accordingly to the radiation with heating or cooling and thus leads to changes in the transport processes in the planetary boundary layer. Wind can be divided in three different types in this layer, the mean wind, the turbulence and waves (Stull,
1988). The mean wind is responsible for fast horizontal wind or advection of meteorological parameters and air pollutants. However, turbulence dominates the vertical transport. Waves, which can often be observed in the nocturnal planetary boundary layer, transport momentum and energy.

In general, the structure of the planetary boundary layer can be divided into different (sub-) layers according to their thermodynamic characteristics. A key parameter for air quality is the vertical depth of the so called mixing layer (ML). The top of
the mixing layer is characterized by a strong gradient in parameters such as potential temperature and aerosol concentration. The top of the mixing layer has a distinct diurnal cycle that depends on both the synoptic and local weather conditions, in particular the energy balance at the surface as well as the local topography. During fair weather days this layer is characterised by an unstable layer and strong mixing due to turbulence during the day. The turbulence in the mixing layer is mainly due to





convection and mixes e.g. heat, moisture and momentum but also air pollutants in the vertical. This part is often also called the convective mixing layer. The mixing layer during daytime is separated from the free atmosphere by an entrainment zone. This is a stable layer which prevents the transport of e.g. air pollutants form the mixing layer to the free atmosphere. During the night a stable layer (also called the stable boundary layer) develops upwards from the ground, capped by a residual layer.

This stable boundary layer is characterised by stable stratification with weak and sporadic turbulence and low mixing. After sunrise and the corresponding increase of the solar radiation, the stable boundary layer and the residual layer dissolve rapid and the mixing layer grows. In case of cloudy and rainy conditions the diurnal evolution of the mixing layer is less pronounced resulting in a lower extent of this layer (Stull, 1988). In spite of its importance there are only indirect methods to determine the mixing layer height (MLH) from measurements taken by remote sensing or radio soundings (e.g. Emeis et al., 2008; Eresmaa

et al., 2006; Ketterer et al., 2014; Wiegner et al., 2014; Emeis et al., 2012).

A ceilometer (Vaisala CL31, Finland) (Münkel, 2007) was deployed to measure vertical profiles of the aerosol attenuated backscatter signals during the SusKat-ABC campaign (December 2012 to June 2013) and beyond until May 2016 at Bode, the supersite of the SusKat-ABC campaign, located in a semi-urban setting in the Kathmandu Valley. A ceilometer is a single-wavelength backscatter lidar, which measures the vertical profile of the aerosol attenuated backscatter signal. More details on

this method are presented in 2.2. These measurements are used to determine the height of the mixing layer. To our knowledge these data are the first multi-year ceilometer measurements in the Kathmandu Valley, and in the greater Nepal region, and provide a unique dataset for the analysis of the meteorological conditions and the air quality in this region. The main aim of setting up the ceilometer instrument as part of the measurement campaign was to better understand the atmospheric dynamics in the valley and to thereby support the analysis of the air pollutant measurements. In this study the mixing layer height data

were used to analyse the diurnal variation of the mixing layer height for each season and its dependency on the meteorological conditions and they provide information on the relation between air pollution concentration and the mixing layer. As an example of such relationships, black carbon (BC) measurements simultaneously measured at the Bode site are analysed and presented. Black carbon is chosen because it has a very low chemical reactivity in the atmosphere and can thus be seen as a tracer for air pollution. Its primary removal process is wet or dry deposition (Bond et al., 2013). This analysis focuses on the

impact of the diurnal variation of the mixing layer height on the temporal variation and the magnitude of the black carbon concentration for each season of the year. This also allows for an assessment of the impact of the mixing layer height on air pollutant concentrations compared to other processes such as the amount and the timing of emissions, horizontal advection and deposition. Furthermore, the information on the diurnal cycle of the mixing layer height and the black carbon concentration were used to estimate the black carbon emission flux in the morning hours for each month.

# 2   Method

## 2.1   The SusKat-ABC measurement campaign

A central part of the SusKat project is a seven month long (December 2012 to June 2013) field campaign (SusKat-ABC) with two month long intensive campaign (December 2012 to February 2013) in Nepal with a focus on the Kathmandu Valley



providing detailed observations of a large number of chemical compounds and meteorological parameters (Rupakheti et al., 2016). The name SusKat-ABC refers to the collaboration with the Atmospheric Brown Cloud (ABC) Programme of the United Nations Environment Programme (UNEP), which provided access to a wide network of researchers in Asia who conducted measurements during the campaign. From December 2012 to June 2013, more than 40 scientists from nine countries represent-

ing 18 research groups deployed more than 160 state of the art scientific measurement instruments for intensive ground-based monitoring. The measurement network includes one urban supersite and five satellite stations within and on the rim of the Kathmandu Valley, and five regional sites in the broader region surrounding the Kathmandu Valley and other parts of Nepal. Some of the instruments deployed in the Kathmandu Valley continued to measure beyond the SusKat-ABC campaign and thus collected long term data that provided a first opportunity for analysis of long-term characteristics of air pollution and

meteorology.

Geographically, Nepal stretches along the Great Himalaya Range and has a diverse topography. Nepal is one of the countries with the highest average elevation and is characterized by a steep north-south gradient in surface elevation. The Kathmandu Valley is a bowl-shaped elevated basin at an altitude of 1300 m (asl). It has a nearly flat bottom with an area of 340 km$^2$. The valley is surrounded by mountain ridges ranging between 2000 m and 2800 m (asl) with five passes (200 – 1500 m above

valley floor) and a river outlet. The valley is particularly vulnerable to air pollution because of its bowl-shaped topography which restricts air flow and ventilation of pollutants.

The meteorology of Nepal is dominated by the Asian monsoon circulations. The basics of the monsoon meteorology are similar each year but the exact timing of the beginning and the end of the individual monsoon periods vary each year. The yearly cycle can be broken down into four basic seasons: the pre-monsoon season (March, April, and May), the monsoon (June, July,

August and September), the dry or post-monsoon season (October, November) and the winter season (December, January, and February) (e.g. Shrestha et al., 2000). The valley's meteorology is influenced by both large scale synoptic features and the local mountain valley circulation. Typically, the valley receives up to 90 % of its annual precipitation during the monsoon months in summer.

In this study the measurements with a Vaisala ceilometer (CL31) are used to study the vertical structure of the atmosphere,

especially the mixing layer height. To cover a whole annual cycle, data from March 2013 to February 2014 are used here. In addition to the ceilometer data also solar radiation, wind, precipitation and black carbon measurements are used in this study. All instruments are located at the Bode site (Fig. 1) in the Kathmandu Valley on the roof top of a building about 15 m above ground. The surroundings of the measurement site are characterised by a mixed residential and agricultural setting in a suburban location with only light traffic and less densely scattered buildings than in the centre of the Kathmandu Valley. An

overview on the availability of the mixing layer height and the black carbon data used in this study is given in Table 1. The meteorological parameters were measured with an automatic weather station (Campbell Scientific, UK) at a time resolution of one minute. Global solar radiation observations were taken with an instrument measuring total sun and sky solar radiation. Its spectral range of 360 to 1120 nm encompasses most of the shortwave radiation that reaches the Earth's surface. The black carbon concentrations were measured by an Aethalometer (Aethalometer AE33, Magee Scientific, USA) (Drinovec et al.,





2015) recording data at a time resolution of one minute. All data are used with a time resolution of one hour calculated as a mean from the original data.

## 2.2 Ceilometer measurements and analytical methods

Ceilometers are routinely used for automatic reports of the mixing layer height. Numerous studies (e.g. Emeis et al., 2008;
Eresmaa et al., 2006; Helmis et al., 2012) show that ceilometer data are suitable for studying the vertical characteristics of the atmosphere and for using the obtained profiles to assess the height of aerosol layers and the mixing layer height. Other techniques used to determine the mixing layer height such as sodar (Sound Detecting And Ranging) or radiosonde measurements show a good agreement with results from ceilometer data also in mountain areas (Ketterer et al., 2014). In Münkel et al. (2011) measurements taken with eye-safe Vaisala ceilometers of the same kind as in the SusKat project were
treated with the same algorithm to determine the mixing layer height as used in this study. They are compared to results from radiosondes obtained during a campaign in Germany. There is a good agreement with mixing layer heights derived from the relative humidity and potential temperature profiles reported by radio soundings and the heights from the ceilometer measurements.

The ceilometer CL31 used in this study employs pulsed diode laser LIDAR technology (LIDAR = Light detection and
ranging), where short laser pulses are sent out in a vertical or near-vertical direction. The operating principle of the ceilometer is based on the measurement of the time needed for the short laser impulse to traverse the atmosphere from the transmitter of the ceilometer to a backscattering volume, e.g. fog, mist, precipitation, clouds, and back to the receiver of the ceilometer. Knowing the speed of light, the time delay between the launch of the laser pulse and the detection of the backscatter signal indicates the cloud base height. The ceilometer CL31 is able to detect three cloud layers simultaneously. Because the laser
light is attenuated by atmospheric particles on its way to the backscattering volume and back to the ceilometer the profile is better called 'attenuated backscatter'. Prior to transmission, the received signal is multiplied by the square of the distance to the backscattering volume and divided by the overlap factor, which accounts for the fraction of the light cone of the transmitter that is within the field-of-view of the receiver. This procedure ensures the comparability of the signal received from different distances, for details see e.g. Kotthaus et al. (2016).

From the attenuated backscatter profiles, data points with observed clouds within the mixing layer or with precipitation are excluded from the analysis. Therefore the degree of signal attenuation is so low that the attenuated backscatter profile can be used as a proxy for the aerosol density in the observed backscattering volume. The algorithm used to determine the aerosol density and mixing layer height from ceilometer data collected in the Kathmandu Valley is the Vaisala BL-VIEW algorithm. It has been introduced in Münkel and Roininen (2010) and is a part of the commercial Vaisala software product BL-VIEW. This
algorithm is based on the gradient method looking for the gradient minima in the attenuated backscatter profiles that mark the top of aerosol layers (Münkel, 2007). This is based on the assumption that within the mixing layer the aerosol concentration is nearly constant in the vertical and distinctly higher than above (Steyn et al., 1999).

In addition to the attenuated backscatter, every profile contains a considerable amount of noise that is generated by the receiver electronics and stray daylight. Because for single profiles, this noise can outweigh all atmospheric structures, a profile





correction is necessary. Thus a vertical and temporal averaging procedure is applied to avoid a detection of false layers gener-
ated by signal noise. The amount of signal noise depends on the range and time of the day, thus the enhanced gradient method
introduces variable averaging parameters. Long averaging intervals help to prevent false gradient minima hits generated by
signal noise. On the other hand, this approach reduces the ability of the algorithm to respond to short scale signal fluctuations
in space and time. In the SusKat project the time averaging interval for the backscatter signal is 15 to 52 minutes depending

5  on the signal noise. The choice of the intervals used for averaging over height is independent of noise; to account for the noise
amplification introduced by the range squared multiplication, the sliding averaging intervals rise gradually from 80 m for all
10 m height bins below 200 m to 360 m for all 10 m height bins above 1500 m. Example: The attenuated backscatter value
at 100 m height is replaced with the average of the values between 60 m and 140 m. Not all gradient minima are reported by
the BL-VIEW algorithm; both signal and gradient have to exceed threshold values that are a function of signal noise. Espe-

cially during daytime situations with small aerosol load, aerosol layer reports are often not available. Small density fluctuations
within a convective boundary layer with a high aerosol load can lead to low gradient values which could be reported as the top
of the aerosol layer. To prevent the report of these fluctuations as the top of the aerosol layer, variable threshold values for the
signal and gradient are introduced. These threshold values are a function of the signal noise and only local gradient minima
below the threshold are reported.

For the evaluation in the frame of the SusKat project, the lowest layer height detected was regarded as the mixing layer
height and the nocturnal stable boundary layer, respectively. Figure 2 shows the colour coded attenuated backscatter profile
averaged following the procedure explained above for 24 h together with the aerosol layers reported by BL-VIEW (rectangles)
and cloud bases reported by the ceilometer (blue and red circles). The yellow and black vertical lines indicate sunrise and
sunset. Only from 1 p.m. till 7 p.m. local time, the mixing layer height exceeds 500 m, during the rest of the day it barely

exceeds 100 m. BL-VIEW reports up to three aerosol layer heights every 10 min. In the scope of the SusKat project, a more
convenient time base for comparison with air quality and meteorological parameters is one hour. Therefore for every hour the
last 6 reported 10 min values are examined to find the most applicable value to represent all 10 min mixing layer height values
of the last hour. For this purpose a weighing procedure is applied to find the height value with most hits in its vicinity. As part
of this weighing procedure a score $S(MLHt_x)$ for every 10 min value per hour ($t_x$) is calculated and the reported 10 min mixing

layer height with the highest score gets reported as the mixing layer height of this particular hour. For the calculation of the
score $S(MLHt_x)$ of the 10 min mixing layer height at time $t_x$ only the 10 min values per hour ($MLHi$, i = 1 to 6 with $|MLHi -
MLHt_x| < 200$ m) are taken into account which are within a distance of less than 200 m from the height value $MLHt_x$.

$$S(MLHt_x) = \sum_{MLHt_x - 200 < MLHi < MLHt_x + 200} 1 - \frac{|MLHi - MLHt_x|}{200} \tag{1}$$

The lowest of the height values $MLHt_x$ with highest score $S(MLHt_x)$ is reported as mixing layer height presenting best all

detections within the past hour. To some extent it resembles a median calculation, but the 200 m distance restriction avoids
unrealistic results in two layer situations like this: if 310 m, 330 m, 250 m, 600 m, 620 m, 620 m are the 10 min values, then
$S(310\,m) = 2.6$, $S(600\,m) = 2.8$, $S(620\,m) = 2.9$, and 620 m gets reported.





The ceilometer deployed in the Kathmandu Valley is a "first generation" ceilometer (CL31). During the time period investigated in this study, it had not been operated with settings fitting best for mixing layer height assessment. For example the report interval was set to 120 s instead of 16 s. Before applying the algorithm to determine the mixing layer height, the attenuated backscatter range has to be converted to attenuated backscatter height with 10 m height resolution. Tilting angle and range resolution is taken into account for this. Here a range resolution of 20 m instead of 10 m was chosen. Nevertheless, most of the

time the aerosol load in the Kathmandu Valley was still sufficiently high for reliable mixing layer height retrievals. See Kotthaus et al. (2016) for a more detailed discussion of the influence of CL31 hardware and firmware versions on the backscatter profiles.

## 3  Results

### 3.1  Seasonal variation of the diurnal cycle of the mixing layer height

The diurnal cycle of the mixing layer height is typically characterized by low mixing layer heights during the night, gradually increasing after sun rise and reaching maximum values in the afternoon before decreasing again later in the day (Fig. 3). Seasonal differences can be seen particularly in the duration of enhanced mixing layer heights during daytime and in the measured minima and maxima. In the pre-monsoon season (Fig. 3a) the diurnal cycle is very distinct with a low mixing layer in the night and morning hours (median around 130 m) and an increase of the height from about 10 a.m. to a maximum between

4 to 6 p.m.. After 6 p.m. the mixing layer height decreases rapidly and the values of the median mixing layer height drop from 1130 m down to 470 m, indicating the development of a stable layer from the surface upwards after sunset. The maximum mixing layer heights measured during the pre-monsoon season are in general the highest of all seasons (maximum median value at 5 p.m. of 1210 m). In contrast to the pre-monsoon season the amplitude of the diurnal cycle measured during the monsoon season is much less distinct (Fig. 3b), with the lowest differences of all seasons between the minimum and maximum

of the cycle (maximum difference of medians: 480 m). Furthermore, among all seasons, the lowest mixing layer height during the day (maximum median 700 m) and the highest during the night and morning hours (median 220 – 370 m) is observed in this season. In the post-monsoon season (Fig. 3c) the mixing layer height during the night and morning hours is still high with a median of 150 to 290 m compared to the pre-monsoon season and the maximum is shifted to earlier hours (around 3 p.m.). The diurnal cycle in winter shows a low mixing layer in the morning and night hours with a median of around 150 m (Fig.

3d). It is also characterised by a short duration of enhanced mixing layer heights during daytime (around eight hours) with maximum heights of on average less than 900 m. The same characteristics are shown by the data for the year 2014 (not shown here) which strongly suggests that the diurnal cycles discussed above are quite typical for the particular seasons in that region.



## 3.2 Mixing layer height and meteorology

The development and characteristics of the mixing layer depend to a large extent on the local meteorological characteristics in
the region. Because of their high impact on the development of the mixing layer, also measurements for solar radiation, wind
speed and precipitation at the Bode station are shown in this analysis.

To a large degree, the characteristics of the diurnal cycle of the mixing layer height as described in 3.1 are consistent with the
characteristics of the diurnal cycle of these meteorological parameters. As a key driving force for convection, solar radiation
has a strong impact on the development of the mixing layer and thus, both diurnal cycles show in general similar structures
(Fig. 4). The maximum of the mixing layer height, however, is shifted a couple of hours towards the late afternoon compared
to the maximum in solar radiation. This is due to a delayed response in the production of thermal turbulence to the warming of
the ground by the incoming solar radiation. In addition to convection also the occurrence of mechanical turbulence in the PBL,
indicated by the wind speed (Fig. 5), has an impact on the formation of the mixing layer height. In all seasons the measured
wind speeds are typically low with a maximum of less than $6\,\mathrm{m\,s^{-1}}$. During night time the wind speed in all seasons is typically
below $1\,\mathrm{m\,s^{-1}}$, which indicates, together with the very low mixing height in the pre-monsoon and winter season, very stable
conditions with only low vertical mixing during the night and morning hours.

The seasonal characteristics of the mixing layer height diurnal cycle are also reflected in the seasonal cycles of the mete-
orological parameters. The highest daily maximum mixing layer is measured in the pre-monsoon season when also the solar
radiation is the strongest (maximum median 1210 m, 828 $\mathrm{W\,m^{-2}}$) (Fig. 4a), whereas both are lowest in the winter season
(maximum median 830 m, 582 $\mathrm{W\,m^{-2}}$) (Fig. 4d). The diurnal cycle of the mixing layer height for the monsoon season was
found to be slightly different from the other seasons. This is also observed in the diurnal cycles of solar radiation and wind
speed. The diurnal cycle of the solar radiation shows a larger range of values per hour of the day given by, for instance, the 25th
and 75th percentiles (Fig. 4b), than in the other seasons. This indicates transition from clear sky conditions to a cloudy regime
and the impact of the more frequent occurrence of cloud layers during this season leading to lower solar radiation values. The
consequential reduction in energy leads to a reduced formation of turbulence and thus to lower mixing layer heights. This can
also be seen when comparing the diurnal cycle averaged over all days (grey) with the cycle obtained when averaging only over
days with precipitation not exceeding a sum of 0.5 mm per day (dry days) (light orange) in Fig. 6. When extracting the mixing
layer height from the ceilometer measurements all hours with clouds or precipitation were not taken into account, in contrast
to that using only 'dry days' means that the whole day with a sum of precipitation exceeding 0.5 mm is omitted. Only 24 %
of the whole dataset of mixing layer height is considered in the calculation of the diurnal cycle for dry days in the monsoon
season (Tab. 1). Figure 6 therefore illustrates the impact of rainy days and the shielding of solar radiation by the associated
clouds on the height of the mixing layer in the monsoon season. Compared to the diurnal cycle of the mixing layer height for all
days, the diurnal cycle for "dry days" is more distinct and shows a much higher maximum mixing layer height during daytime
(difference of max. median around 240 m). This difference between dry and all-day mixing layer heights is much smaller in
the other seasons. Therefore the presence of clouds and rain in the monsoon season explains a large part of the on average
lower mixing layer height and higher variability compared with other seasons.





The described differences in the mixing layer height between the seasons during night time are consistent with the seasonal variation of the surface temperature and the heat capacity of the soil. Higher average surface temperatures in the monsoon and post-monsoon seasons lead to higher mixing layer heights during the night compared with pre-monsoon and winter conditions. Another potential reason is the presence of clouds during the night, especially in the monsoon season, which reduces the outgoing longwave radiation and thus the related radiative cooling of the surface.

In addition to the impact of the meteorology on the formation of the mixing layer in mountain regions, the topography and in this case more specifically the structure of the mountain valley is also important. Panday and Prinn (2009) and Panday et al. (2009) studied the dynamics of the basin's nocturnal cold air pool, its dissipation in the morning, and the growth and decay of the mixing layer over the Kathmandu Valley during daytime in the dry season. These observations and additional model studies suggested that the cold pool dissipation was dominated by upslope flows along the valley rim accompanied by subsidence over the basin centre. In addition, during fog free days there was erosion of the inversion from below by the growth of thermals forming over a warming surface. They also found that the Kathmandu Valley's mixed layer height peaks around noon, and then decreases again over the course of the afternoon, which is consistent with the findings of this study.

### 3.3 Black carbon concentration and the mixing layer height

#### 3.3.1 Impact of the mixing layer on black carbon concentration

In this section the impact of the mixing layer height on the concentration of black carbon is analysed and compared qualitatively to the impact of other important processes such as emissions, meteorological conditions and dry and wet deposition. The vertical structure of the mixing layer is important for the concentrations of black carbon at the surface due to its impact on the volume into which pollutants are mixed, the strength of the vertical mixing and for the transport of pollutants into the residual layer in the evening and out of this layer into the mixing layer in the morning.

The ventilation coefficient, calculated as the product of the mixing layer height and the 10-m wind speed, provides information on the intensity of the transport and mixing of pollutants within the mixing layer (Tang et al., 2015). For this study wind speed measurements are only available at 15 m above the surface, thus the calculated ventilation coefficients are expected to be slightly higher than when using the 10-m wind speed. The values of the ventilation coefficient at the Bode site are highest in the pre-monsoon and post-monsoon season (Fig. 7) with a maximum median of up to $5200 \, \mathrm{m^2 \, s^{-1}}$. In the monsoon and winter season the median values does not exceed $2000 \, \mathrm{m^2 \, s^{-1}}$ and $3000 \, \mathrm{m^2 \, s^{-1}}$, respectively. During night time the ventilation coefficients show a small variation with a median minimum of only 43 and $105 \, \mathrm{m^2 \, s^{-1}}$ depending on the season. This suggests that during the night, stable stratification and the consequential low mixing and the transport of air pollutants are very small potentially resulting in a strong accumulation within the mixing layer. Median daytime values of the ventilation coefficient above $1000 \, \mathrm{m^2 \, s^{-1}}$ occur only between four (monsoon and winter) and eight pre-(monsoon) hours per day. Pollutants are thus mixed and transported only during relatively short periods. These low ventilation coefficients during night time indicate that local emission sources play an important role in the high air pollution levels observed in the valley.





For the analysis of the impact of the mixing layer height on the black carbon concentration in Fig. 8 the measured mixing layer heights were grouped into classes in such a way that the impact of other meteorological drivers is minimized (Wagner, 2014). The optimum size of the classes (C) is different for each season and is approximated using the approach of Sturges (1926) relating the range of values (R = MLHmax – MLHmin) and the total number of values (n) as follows:

$$C = \frac{R}{(1 + 3.322 \times lg(n))} \tag{2}$$

For every mixing layer height class the median of the corresponding black carbon measurements is shown by the height of the bars in Fig. 8. The widths of the bars indicate the number of black carbon measurements available per class. In general there is a clear anti-correlation of the black carbon concentration and the mixing layer height in all seasons except for the monsoon season. The strength of the decrease of black carbon concentration with increasing mixing layer height, however, differs substantially between the seasons. In general, black carbon concentrations are high when the mixing layer height is low and vice versa. This is mostly due to the larger volume into which pollutants can be vertically mixed and the associated stronger turbulence (and thus mixing) when the mixing layer is high. The described relation between the two parameters is also illustrated in the diurnal cycle of the black carbon concentration and the mixing layer height shown in Fig. 9. In all seasons the black carbon concentration shows a minimum during the day when the mixing layer height is at its maximum.

In the pre-monsoon season the highest black carbon concentrations (around 30 $\mu g\,m^{-3}$) are measured when the mixing layer is below about 200 m (Fig. 8a) as it is the case during the night (Fig. 9a). Mixing layer heights above 200 m are mostly measured during daytime (11 a.m. – 11 p.m.) when the median concentrations are low (around 5 $\mu g\,m^{-3}$). After about 6 p.m. the decrease in mixing layer height goes along with an increase in the black carbon concentration which is caused to a large extent by the decreasing height and thus decreasing volume of the mixing layer (Fig. 8a). The maximum black carbon concentrations coincide with the minimum of the mixing layer height in the morning hours. The relationship between the black carbon concentration and the mixing layer height is similar in winter and the pre-monsoon season (Fig. 9a and d).

In the monsoon season the correlation between the black carbon concentration and the mixing layer height is not as strong as in the other seasons (Fig. 8b). The black carbon concentrations are high for mixing layer heights below 200 m but the decrease in black carbon concentration with the increase of the mixing layer height is less pronounced and for very large mixing layer heights the black carbon concentrations even increase. This suggests that other processes than the temporal evolution of the mixing layer height such as the temporal variation of local emissions strengths are also of importance here. In contrast to the pre-monsoon and winter season, the diurnal cycle of the black carbon concentration in the monsoon season shows a second maximum in the evening hours (around 8 p.m.) (Fig. 9b). The increase in the concentration in the afternoon is stronger than in the pre-monsoon season. But despite a further decrease of the mixing layer height the black carbon concentration decreases again in the evening and remains on a nearly constant level during the night until it increases again in the early morning towards the daily maximum in the morning (at around 8 a.m.). These main characteristics of the black carbon diurnal cycle remain the same when only considering dry days (Fig. 10). This suggests that more frequent precipitation events and thus wet deposition is not the main reason for this decrease in black carbon concentrations after 8 p.m.. The diurnal cycle of black carbon in





the post-monsoon season shows a similar shape as in the monsoon season although the second peak in the afternoon is less pronounced and the concentrations are somewhat higher.

In Putero et al. (2015) black carbon measurements at an urban site located in the west of the Kathmandu Valley are described and discussed. In all seasons a diurnal cycle of black carbon typical for an urban site with two daily maxima, one prominent peak in the morning (between 7 a.m. and 8 a.m.) and a second peak in the evening (between 8 p.m. and 9 p.m.) as well as minima at night (between 1 a.m. and 2 a.m.) and in the afternoon (between 2 p.m. and 3 p.m.) was found. Panday and Prinn (2009) observed such a diurnal cycle also for carbon monoxide (CO) and particulate matter ($PM_{10}$). Their analysis based on

observations shows that this double peak results from an interplay between the timing of the emissions and ventilation.

    Putero et al. (2015) argue that the first daily black carbon peak can be explained by increased emissions from e.g. traffic and cooking activities under stable atmospheric conditions and a low mixing layer height. Over the course of the day, the valley is ventilated by westerly winds entering through the western passes, the dilution within the higher PBL and decrease of emissions can explain the daily minimum in black carbon (Putero et al., 2015). These characteristics (westerly wind, highest wind speeds

during daytime, highest mixing layer height) are also found at the Bode station as described above. When the PBL height starts to decrease due to the decrease in solar radiation and surface heating along with the higher emissions of evening traffic and cooking activities, a secondary peak in black carbon is observed in Putero et al. (2015). The authors argue that the following decrease in black carbon concentrations is caused by the decrease in traffic and domestic emissions and a less efficient vertical mixing within a more stable mixing layer. But a less efficient vertical mixing would rather lead to increased or constant level of pollutants. In general the above mentioned findings and explanations are most likely also valid for the diurnal black carbon

concentrations at the Bode station in the monsoon and post-monsoon season as it shows the same main characteristics. In contrast, the diurnal cycle of the black carbon concentration observed at the Bode station during the pre-monsoon and winter season shows one maximum in the morning hours.

    The anti-correlation of the one year time series of the black carbon concentration and the mixing layer height of -0.54

(pearson coefficient) or the pre-monsoon season, -0.19 for the monsoon season, -0.40 for the post-monsoon season and -0.46 for the winter season shows that a part of the variation in time of black carbon can be explained by atmospheric dynamics and that its magnitude depends on the season. Other processes important to explain the variability of the level of black carbon are temporal variation of emissions as well as large-scale transport and deposition. For the pre-monsoon season also the anti-correlation of the diurnal cycle of the black carbon concentration and the mixing layer height as well as of the wind speed

in the morning and daytime in the pre-monsoon season suggests that a large part of the black carbon diurnal cycle is driven by atmospheric dynamics. The missing second peak in the evening indicates that other processes than described above are dominating here. Because differences between the black carbon diurnal cycles in the different seasons were not found in Putero et al. (2015) the exact location of the station also seems to be an important factor. Putero et al. (2015) argue that the diurnal cycle of the black carbon concentration indicates that local pollution sources, mostly related to road traffic or domestic

emissions, represent the main contribution to air pollution in the valley. The Bode site is not located directly in the urban core area but more towards the rather suburban east. Pollution roses show that between about December and April most of the highest black carbon concentrations coincide with wind from the east and east south east. This is where a number of brick kilns




are located which operate mainly during this time of the year. The brick kilns emit continuously throughout the day whereas in the other months sources with more pronounced diurnal cycles, such as traffic and cooking activities, are dominating the total

black carbon emissions. Thus, in addition to the circulation in the valley, the location and the main emission sources as well as the time profiles of these sources have a great impact on the black carbon concentrations as well as their diurnal cycle in each season.

Highest black carbon concentrations are measured during the pre-monsoon and winter season (Fig. 9a and d), which was also found by Putero et al. (2015). Several processes can lead to these high concentrations. In the winter season the low mixing layer height and wind speed favour the accumulation of pollutants due to the limited transport and vertical mixing. But in the pre-monsoon season, high black carbon concentrations are observed, despite the comparable high ventilation coefficient (Fig.

7a). Additional emission sources such as domestic heating and brick kilns as well as the lack of precipitation are expected to be the main factors leading to the high black carbon concentrations during these seasons. In the monsoon season wet deposition of black carbon by precipitation leads to a lower level of black carbon concentration in the atmosphere.

### 3.3.2   Estimation of black carbon emission fluxes

The annual mean emission estimates from EDGAR HTAP inventory v2.2 (Janssens-Maenhout et al., 2000) for the Kathmandu

region range between $28\,\mathrm{ng\,m^{-2}\,s^{-1}}$ in winter and $19\,\mathrm{ng\,m^{-2}\,s^{-1}}$ in rest of the year (annual mean $= 21\,\mathrm{ng\,m^{-2}\,s^{-1}}$). EDGAR HTAP has a resolution of $0.1°$ and is based in this region on data from the REAS emission database (Kurokawa et al., 2013) which has a resolution of $0.25°$. First test simulations (not shown) with the WRF-Chem model (Grell et al., 2004; Fast et al., 2006) using this emission inventory underestimate the observed black carbon at the Bode station by a factor of about eight in February 2013. (More information on the model set-up used for these simulations is presented in the supplement material.)

The measured monthly mean black carbon concentrations at the Bode station are in the range of 3.4 and $25.9\,\mathrm{\mu g\,m^{-3}}$ between March 2013 and February 2014. Black carbon concentrations in a similar range were measured for example in Delhi and Mumbai. In Tiwari et al. (2015) monthly mean concentrations from 6.4 to $20.3\,\mathrm{\mu g\,m^{-3}}$ averaged over five stations in Delhi were measured between September 2010 to January 2011. In Mumbai measured black carbon levels during winter season were from 3.2 to $9.4\,\mathrm{\mu g\,m^{-3}}$, compared to 1.0 to $6.9\,\mathrm{\mu g\,m^{-3}}$ in summer and 1.0 to $5.8\,\mathrm{\mu g\,m^{-3}}$ in the monsoon season were

found (Sandeep et al., 2013). For these two cities the EDGAR HTAP emission data base gives annual mean emission fluxes for black carbon of $169\,\mathrm{ng\,m^{-2}\,s^{-1}}$ for Delhi and $600\,\mathrm{ng\,m^{-2}\,s^{-1}}$ for Mumbai. The underestimation of the measured black carbon concentration at the Bode station with the model, their similarity to measurements in Delhi and Mumbai, and the higher emission flux in the EDGAR HTAP database for Delhi and Mumbai suggests that the black carbon emission estimates might be underestimated for the Kathmandu region. This also appears to apply to other current emission inventories such as INTEX-B

(Zhang et al., 2009), with an emissions value of $21.1\,\mathrm{ng\,m^{-2}\,s^{-1}}$ for the Kathmandu Valley.

In this part of the study an estimate of the black carbon emission flux for the morning hours is presented using the observed mixing layer height and black carbon concentrations at the Bode site. The method proposed here to obtain an estimate of the black carbon emission flux is mainly based on three assumptions:



- black carbon aerosols are horizontally and vertically well mixed within the mixing layer,

- the variation of the mixing layer height is only small at night as frequently observed in the measurements (Fig. 3),

- the horizontal transport of air pollutants into and out of the valley and the vertical mixing of pollutants between the mixing layer and the free atmosphere is assumed to be negligible

This means that the black carbon concentration measured at the Bode station is assumed to represent the black carbon load in the column around the station and throughout the valley with the top corresponding to the height of the mixing layer, and thus the product of the measured black carbon and the mixing layer height represents the black carbon mass per unit area within this column. The the third assumption is based on the observation that the wind speed is typically very low ($< 1\,\mathrm{m\,s^{-1}}$) during the night (Fig. 5), and thus the horizontal transport of air pollutants into and out of the valley is assumed to be negligible. And

the mixing of pollutants between the mixing layer and the free atmosphere (entrainment) occurs mostly when the mixing layer height increases or decreases significantly, which is not the case during the hours used for this calculation. These observations allow the conclusion that the main process driving the increase of the black carbon concentration during the night is not the variation in the mixing layer height (Fig. 9), nor mixing of even more polluted air from above or from outside the valley, but rather, as assumed here, the emissions. Assuming that the air in the free troposphere and outside the valley is actually

less polluted than in the highly-populated valley, then any mixing would require a higher emissions rate to compensate the reduction in pollution levels due to mixing, and thus neglecting transport and mixing implies that we will tend to underestimate the actual emissions in the morning hours. Dry deposition is expected to have a negligible effect on the observed mass densities during the course of a single night, given the residence time for fine black carbon particles of approximately six days (Cape et al., 2012), which implies a loss of only approximately 1/144 of the black carbon concentration per hour (i.e., in the range

of a few $\mathrm{ng\,m^{-2}\,s^{-1}}$ for the night time measurements period). Thus not accounting for dry deposition also implies a small underestimate of the emissions. Wet deposition is a very efficient removal mechanism for black carbon. Since we do not account for wet deposition in the calculation, this will also imply that the estimated emission fluxes are an underestimate, especially for months with high precipitation rates. With these three factors together, we can consider our emissions estimate to likely be a lower bound to the actual emissions rate in the morning hours.

The difference between the maximum and minimum black carbon concentration ($\Delta BC = BC(t_y) - BC\,(t_x)$) in the night ($t_x$) and morning hours ($t_y$), represents the increase in the black carbon concentration during this time period. $t_x$ refers to the hour with the black carbon concentration minimum in the night and $t_y$ to the hour with the black carbon concentration maximum in the morning hours. If the mixing layer height were constant, and there were no vertical or lateral mixing into the valley, then the product of $\Delta BC$ and the mixing layer height (MLH) divided by the time difference $\Delta t$ ($\Delta t = t_y - t_x$) would yield the average

emission flux $F_{BC}(t_x,t_y)$ in this time period, i.e., the additional black carbon mass per unit area which is necessary to balance the observed increase in black carbon concentration between the night ($t_x$) and morning hour ($t_y$). However, the mixing layer height is not constant. Since the relative variation during the chosen period is small, a first order estimate can be obtained by using the average mixing layer height (ave(MLH($t_x$), MLH($t_y$)) between $t_x$ and $t_y$ in the calculation. To additionally account for the small change (in general a decrease) in the mixing layer height between the night and the morning hours, a 'mixing layer



collapse factor' needs to be introduced (MLH($t_y$)/MLH($t_x$)). This factor can be understood by considering that if there were no mixing with the atmosphere outside the valley during these hours, and deposition were negligible, then the black carbon mass would be preserved within the mixing layer, and the concentration would increase accordingly as the mass is compressed into a smaller volume. These terms are summarized in Eq. 3, which is used to estimate the black carbon emission flux $F_{BC}(t_x,t_y)$ for the morning hours per month at the station. The equation as written also accounts for the conversion from μg to ng (factor 1000) and from hours to seconds (factor 3600).

$$F_{BC}(t_x,t_y) = \frac{\Delta BC \times 1000 \times ave(MLH(t_x), MLH(t_y))}{\Delta t \times 3600} \times \frac{MLH(t_y)}{MLH(t_x)} \qquad (3)$$

with $F_{BC}(t_x,t_y)$ the black carbon emission flux between time $t_x$ and $t_y$ in $\mathrm{ng\,m^{-2}\,s^{-1}}$, $\Delta$ BC the black carbon concentration

difference between $t_y$ and $t_x$ in $\mathrm{\mu g\,m^{-3}}$, ave(MLH($t_x$), MLH($t_y$)) the average of the mixing layer height between $t_x$ and $t_y$ in m, $\Delta t = t_y - t_x$ in h and MLH($t_y$)/MLH($t_x$) the 'mixing layer collapse factor'. The emission fluxes are calculated based on the mean diurnal cycles per month; $t_x$ and $t_y$ are chosen such that they represent the time with the minimum ($t_x$) and maximum ($t_y$) black carbon concentration in the night and morning. These points in time are quite different for the individual months with a range of 1 a.m. to 5 a.m. for $t_x$ and 8 a.m. to 9 a.m. for $t_y$. As noted above, this approach will tend to underestimate the actual

emissions rate in the morning hours, if either non-negligible amounts of cleaner air are mixed in from outside the valley, or if the deposition loss were much larger than expected based on the literature. Results from this calculation including the results from the different terms in Eq. 3 are given in Tab. 2 and 3 for the four seasons and in Fig. 11 for every month.

The black carbon emission flux estimates for the morning hours obtained with this method (Fig. 11) show a clear seasonal cycle with higher values in the months December to April ($196 - 298 \mathrm{\ ng\,m^{-2}\,s^{-1}}$) than in the other months (26 - 170

$\mathrm{ng\,m^{-2}\,s^{-1}}$). These black carbon emission fluxes are of a similar order of magnitude to the ones from the EDGAR HTAP emission data base especially for highly-polluted Indian city like Delhi, as noted previously. The seasonal variation computed for the emission flux estimates fit well to known seasonal characteristics of the main emission sources. In particular, brick kilns are known to be major emitters of black carbon in the Kathmandu Valley (Chen et al., 2016), and are only active from late December to April. The brick kilns typically emit continuously throughout the day and night, whereas in the other months

sources with more pronounced diurnal cycles, such as traffic and cooking activities, generally dominate the total emissions. In order to consider the daily variability of the estimated emission flux for each month, values based on the 25th and 75th percentile of the diurnal cycles are also shown (Fig. 11).

The emission estimates computed here are subject to further uncertainties in addition to those noted above. One of these uncertainties is the assumption that the measurements at the Bode site are representative for the whole atmospheric column and

the surrounding region, which is difficult to verify without more extensive simultaneous measurements throughout the valley. Furthermore, while the diurnal cycles of black carbon during January to May and November to December have a relatively clear minimum and maximum in night and morning hours, the diurnal cycles of the months June to October show a much higher variability. This makes it more difficult to choose the hour with the minimum and maximum black carbon concentration for the estimation of the flux, and thus adds another uncertainty. On the other hand, the differences between the 25th and 75th





percentile for these periods are relatively small, indicating a greater consistency in the diurnal cycle from day to day in June to October than in the other months.

As a final uncertainty to consider, the possible impacts of wet deposition is not included, which would also cause to tend to generally underestimate the emissions rate if black carbon is being lost during the day due to wet deposition (similar to the effect of dry deposition noted above). While we cannot directly calculate the effect of wet deposition, we can filter out all

days with a sum of precipitation of more than 0.5 mm per day, thus only taking dry days into account in the flux calculation (Table 3). As could be expected, using only dry days generally leads to higher calculated emission fluxes in most of the months compared to considering all days. The differences of the estimated emission flux between all days and dry days range between -9 and 37 $\mathrm{ng\,m^{-2}\,s^{-1}}$. In June, August, September and February the differences are negative, which is likely to be due to the lack of sufficient data to get good statistics for the dry days (e.g. 24 % of the time in the monsoon season for black carbon,

respectively); this may also be influenced by other confounding factors, such as differences in mixing layer characteristics, vertical mixing and horizontal wind speeds on wet versus dry days. Furthermore, precipitation in other parts of the valley, which will generally also have an impact on the measured black carbon concentration at the Bode station, is not taken into account by this approach.

Finally, the estimated black carbon emission fluxes are only valid for the morning hours for which the calculation in Eq. 3

is applied. The implicit assumption in Fig. 11 is that the emission rate during this period is representative of the rest of the 24-hour period as well. This assumption may not be valid; for example, the morning period generally includes the emissions peak from the morning rush hour traffic. While we cannot compute exactly what the 24-hour emissions are based on this technique, without currently unavailable information about the diurnal profiles of all main black carbon sources, it is possible to use the calculation to determine an absolute lower bound on the local emissions, by assuming that the emission rate during all hours

of the day which are not covered in Eq. 3 is zero:

$$F_{BCmin} = F_{BC} \times \frac{\Delta t}{24} \tag{4}$$

The results of this calculation (Table 3 and Fig. 11) show for all months except the monsoon season (June until August), even this absolute minimum for the 24-hour estimated emission fluxes are still higher than the values in the EDGAR HTAP database (28 $\mathrm{ng\,m^{-2}\,s^{-1}}$ in winter and 19 $\mathrm{ng\,m^{-2}\,s^{-1}}$ during rest of the year, as noted above). Thus, even if our analysis

may be biased towards overestimating the 24-hour emissions by including the morning rush hour and other activities (e.g., cooking), they nevertheless provide strong evidence that the emission fluxes in the EDGAR HTAP emission inventory and the other inventories noted above are likely too low for this region

## 4   Summary and outlook

The data of mixing layer height derived from ceilometer measurements analysed in this study represent a unique dataset and

important information for the analysis of the meteorological and air quality conditions in the Kathmandu region. The time




series and especially the diurnal cycles of the mixing layer height show different characteristics in the four seasons. In all seasons the diurnal cycle shows low mixing heights during the night and increasing and maximum values during the day but with seasonal differences in the absolute values of the mixing layer height and the duration of the daytime maximum. Different meteorological parameters such as solar radiation, wind speed and precipitation have been shown to be of key importance

to explain these differences between the seasons. During the monsoon season the observed diurnal cycle of the mixing layer height deviates from that obtained for other seasons by showing the lowest amplitude and the lowest mixing layer height during the day and the highest in the night and morning hours of all seasons. These characteristics can mainly be explained with frequently present clouds and the associated reduction in incoming solar radiation and outgoing longwave radiation.

In general there is a clear anti-correlation of the black carbon concentration and the mixing layer height although this relation is less pronounced in the monsoon season. The daily evolution of the black carbon diurnal cycle differs between the

5   seasons, partly due to the different meteorological conditions including the mixing layer height. Another important reason is the different main emission sources and their diurnal emission profiles relevant in the individual season. From January to April for example brick kilns, which emit continuously throughout the whole day, are major emitters of black carbon, whereas in the other months sources with more pronounced diurnal cycles are dominating the total emissions.

Using the mixing layer height data and the black carbon measurements an estimation of the black carbon emission flux for

10   the night and morning hours was calculated. Even though these estimates are subject to a range of uncertainties, they give a first top down estimate of the order of magnitude of black carbon emissions in the Kathmandu Valley based solely on measurements. The estimated values show a clear seasonal cycle with high values for the months December to April. Compared to the emission flux values provided by different emission databases for this region, the values estimated here are considerably higher. Even accounting for a lower bound for the estimated minimum black carbon emission flux values by assuming no emissions outside the nighttime and morning hours used for the calculation still yields values which are higher than in the emissions databases for all seasons except the monsoon season.

*Acknowledgements.* We would like to acknowledge Shyam Newar, Bhogendra Kathayat and Ravi Ram Pokhrel for their support in operating

the ceilometer and Aethalometer at the Bode site, and Arnico Panday and his colleagues from ICIMOD for logistical support to deployment of instruments and running the Bode site. Financial support for this study was provided by the German Research Foundation (DFG), the Federal Ministry of Education and Research of Germany (BMBF) and the Ministry for Science, Research and Culture of the State of Brandenburg (MWFK). We would also like to thank Ashish Singh for processing the measurment data.



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





**Table 1.** Overview on the availability of the hourly mixing layer height and black carbon data for four seasons during the period March 2013 to February 2014.

| Season 2013/2014 | Total data | MLH data | MLH data only dry days | BC data | BC data only dry days |
|---|---|---|---|---|---|
| March - May 2013 | 2208 | 1579 (72 %) | 1106 (50 %) | 2138 (97 %) | 1517 (69%) |
| June - Sep. 2013 | 2928 | 2052 (70 %) | 716 (24 %) | 2155 (74 %) | 705 (24%) |
| Oct. - Nov. 2013 | 1464 | 1218 (83 %) | 1143 (78 %) | 1345 (92%) | 1212 (83%) |
| Dec. 2013 - Feb. 2014 | 2160 | 2010 (93 %) | 1925 (89%) | 1615 (75%) | 1508 (70%) |

**Table 2.** Terms for the estimation of the black carbon emission flux as used in Eq. 3 and estimated black carbon emission flux for four seasons during the period March 2013 to February 2014.

| Season 2013/2014 | $t_x$ and $t_y$ (local time) | $\text{MLH}(t_x)/\text{MLH}(t_y)$ | average($\text{MLH}(t_x)$,$\text{MLH}(t_y)$) [m] | $\triangle$ BC [$\mu g\, m^{-3}$] | BC emission flux $F_{BC}$ [$ng\, m^{-2}\, s^{-1}$] |
|---|---|---|---|---|---|
| March - May 2013 | 4, 9 a.m. | 1.21 | 164.30 | 17.05 | 188 |
| June - Sep. 2013 | 5, 9 a.m. | 1.36 | 288.00 | 2.34 | 64 |
| Oct. - Nov. 2013 | 5, 9 a.m. | 1.41 | 244.22 | 4.98 | 118 |
| Dec. 2013 - Feb. 2014 | 5, 10 a.m. | 1.33 | 173.48 | 19.46 | 248 |

**Table 3.** Estimated black carbon emission fluxes for four seasons for dry days $F_{BCdrydays}$ and $F_{BCmin}$ as estimated in Eq. 4.

| Season 2013/2014 | BC emission flux dry days $F_{BCdrydays}$ [$ng\, m^{-2}\, s^{-1}$] | BC emission flux estimation of $F_{BCmin}$ (Eq. 4) [$ng\, m^{-2}\, s^{-1}$] |
|---|---|---|
| March - May 2013 | 217 | 39 |
| June - Sep. 2013 | 88 | 11 |
| Oct. - Nov. 2013 | 123 | 20 |
| Dec. 2013 - Feb. 2014 | 254 | 52 |



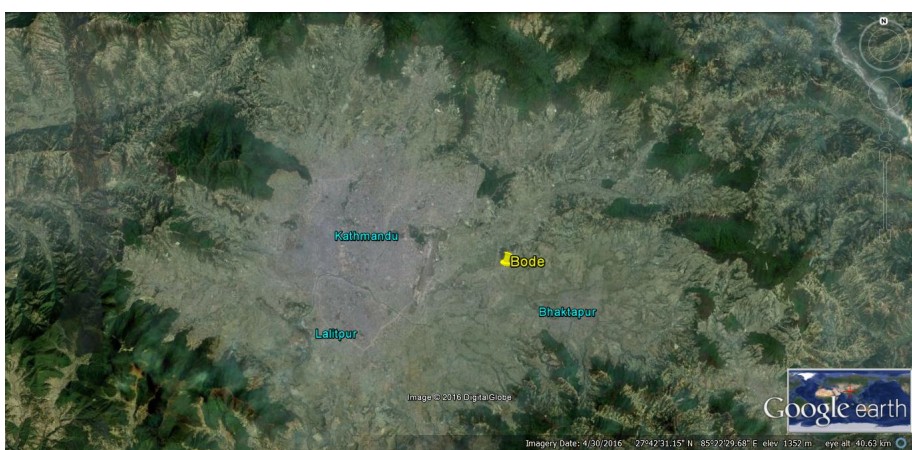

**Figure 1.** Location of the measurement site Bode.

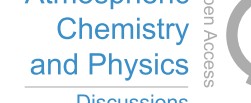



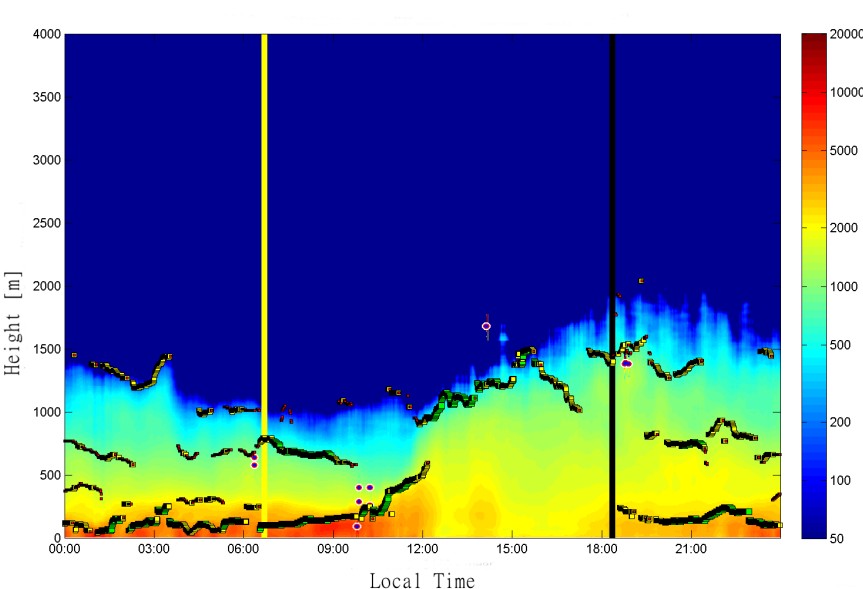

**Figure 2.** Attenuated backscatter density measured on 9.3.2014 $[10^{-9}\,\mathrm{m}^{-1}\,\mathrm{sr}^{-1}]$ together with the aerosol layers reported by BL-VIEW (rectangles) and cloud bases reported by the ceilometer (blue and red circles). The yellow and black vertical lines indicate sunrise and sunset.





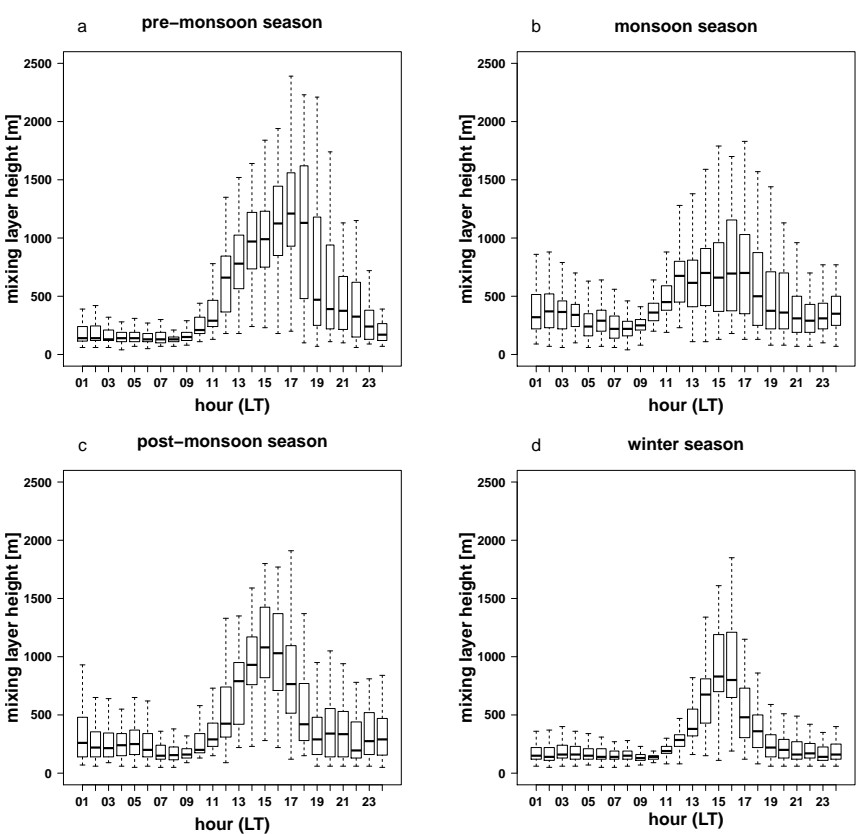

**Figure 3.** Diurnal cycle of the mixing layer height as a Box-Whisker-Plot (showing the median, the upper and lower quantile and whisker) for the (a) pre-monsoon (March – May 2013), (b) the monsoon (June – Sept. 2013), (c) the post-monsoon (Oct. – Nov. 2013) and (d) the winter (Dec. 2013 – Feb. 2014) season at the Bode site.





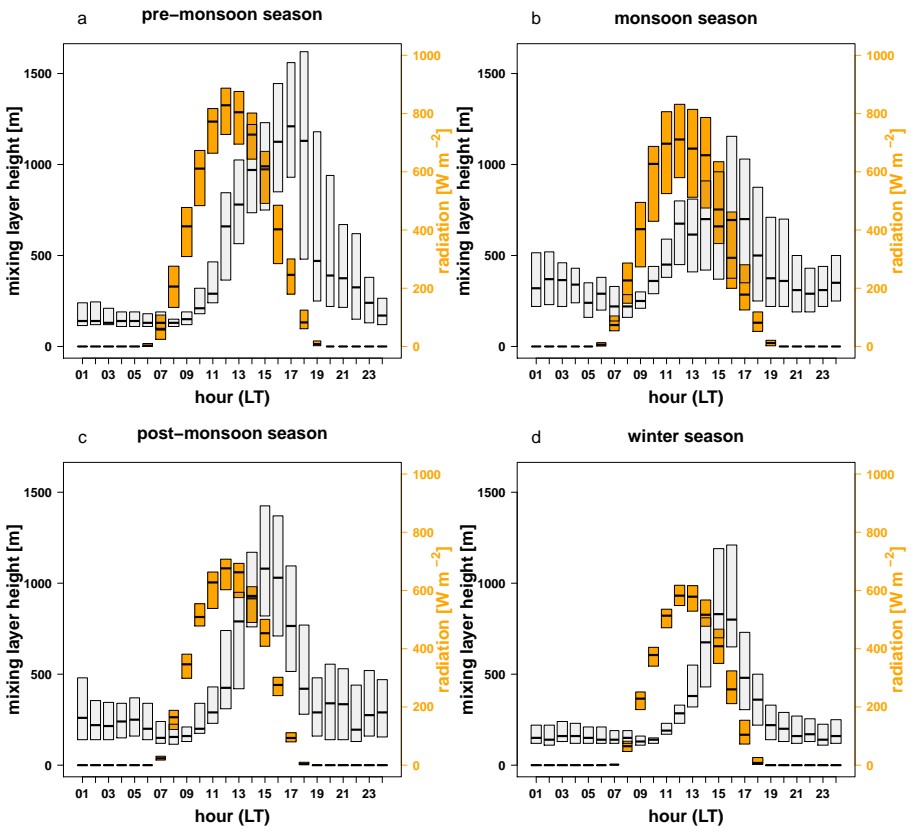

**Figure 4.** Diurnal cycle of the mixing layer height (grey) and solar radiation at the surface (orange) as a Box-Plot (showing the median, the upper and lower quantile) for (a) the pre-monsoon (March – May 2013), (b) the monsoon (June – Sept. 2013), (c) the post-monsoon (Oct. – Nov. 2013) and (d) the winter (Dec. 2013 – Feb. 2014) season at the Bode site.





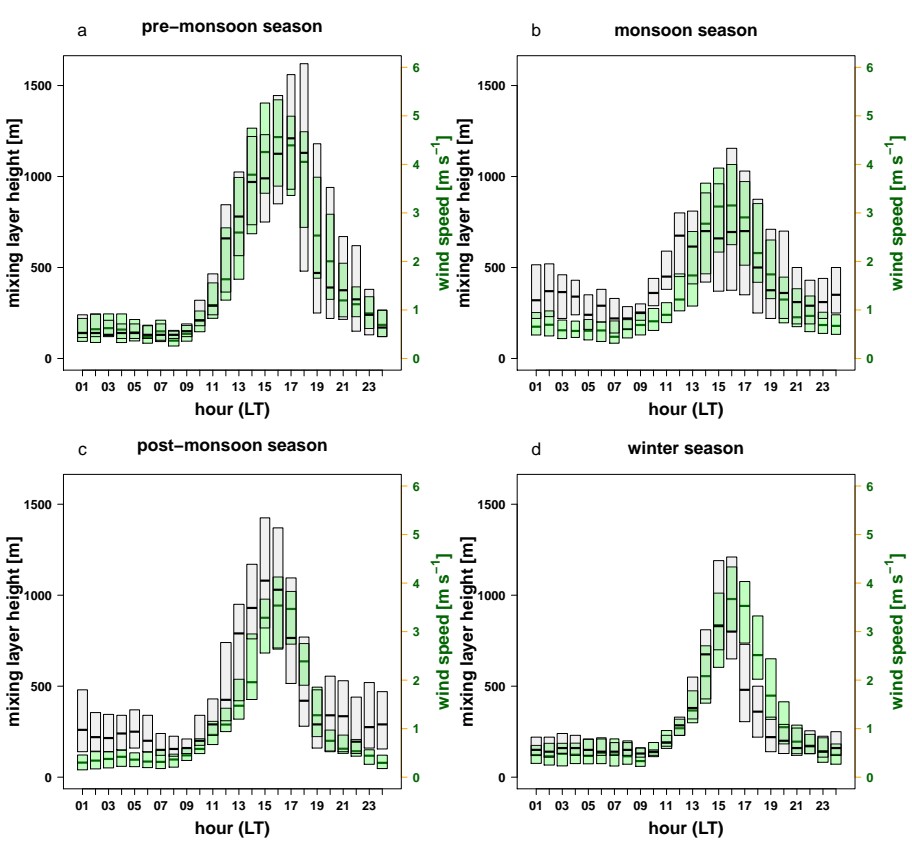

**Figure 5.** Mean diurnal cycle of the mixing layer height (grey) and wind speed (green) as a Box-Plot (showing the median, the upper and lower quantile) for (a) the pre-monsoon (March – May 2013), (b) the monsoon (June – Sept. 2013), (c) the post-monsoon (Oct. – Nov. 2013) and (d) the winter (Dec. 2013 – Feb. 2014) season at the Bode station.





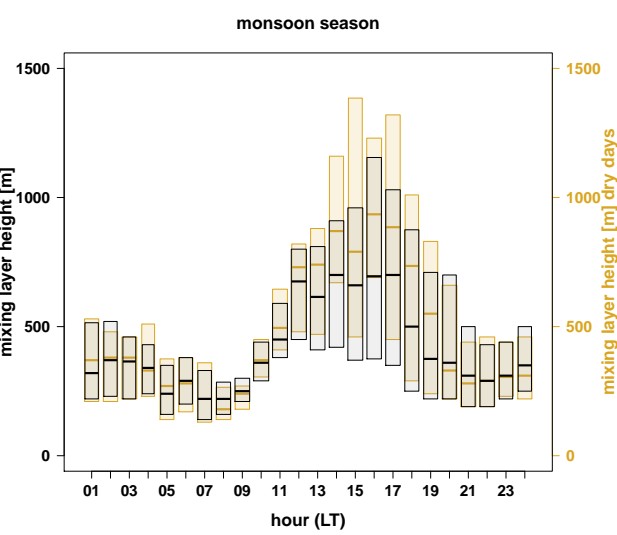

**Figure 6.** Diurnal cycle of the mixing layer height with all days (grey) and only dry days (light orange) as a Box-Plot (showing the median, the upper and lower quantile) for the monsoon season (June–Sept. 2013) at the Bode station.





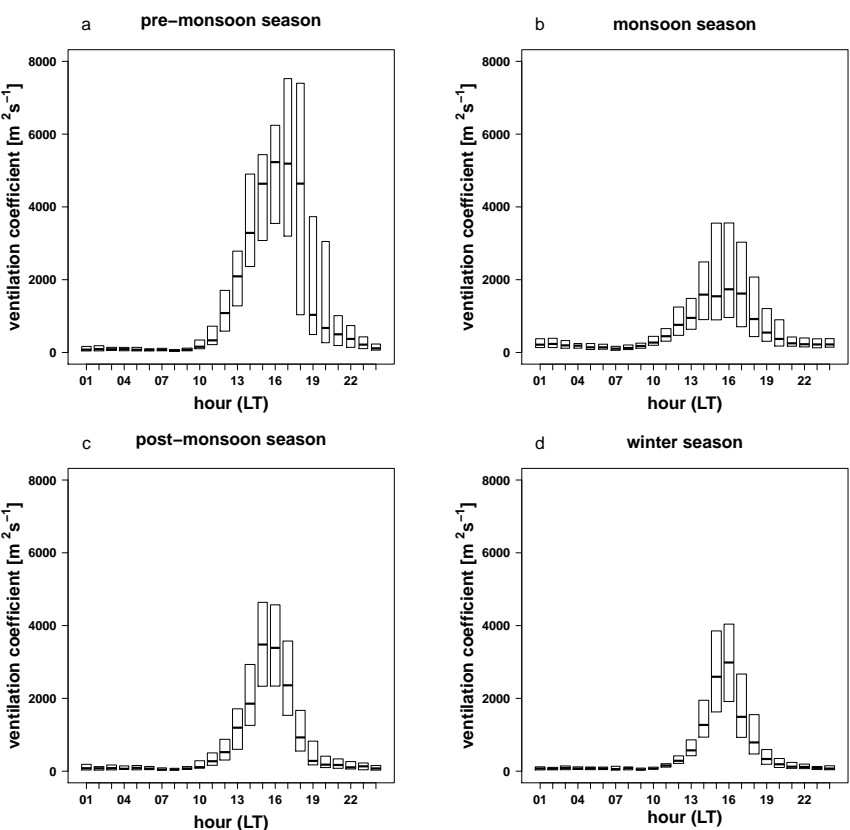

**Figure 7.** Diurnal cycle of the ventilation coefficient as a Box-Plot (showing the median, the upper and lower quantile) for the (a) pre-monsoon (March – May 2013), (b) the monsoon (June – Sept. 2013), (c) the post-monsoon (Oct. – Nov. 2013) and (d) the winter (Dec. 2013 – Feb. 2014) season at the Bode site.





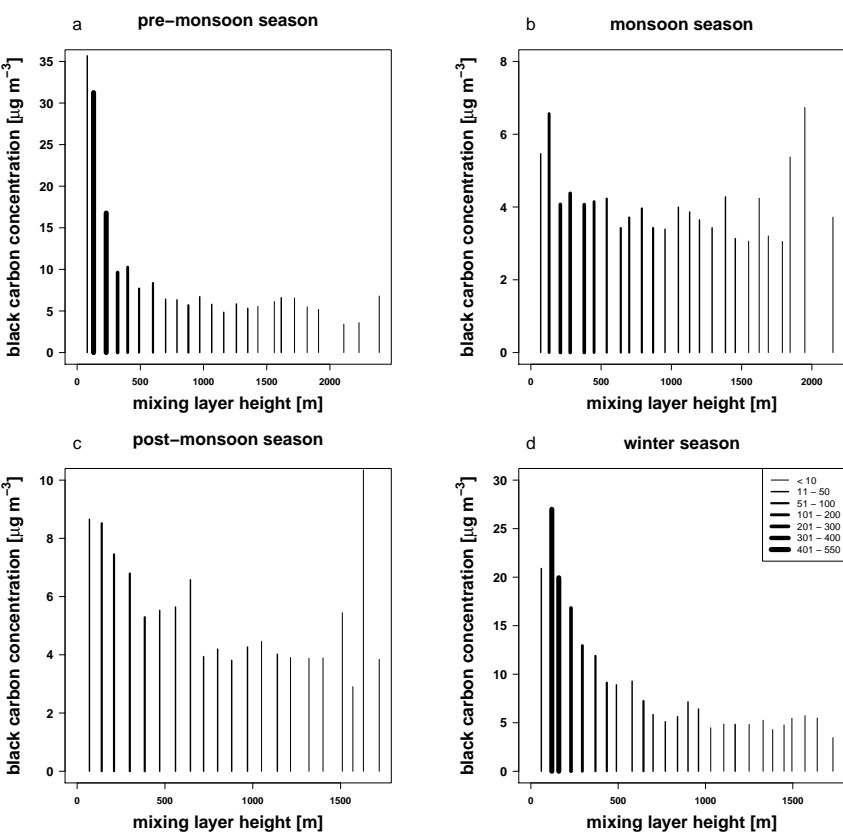

**Figure 8.** Dependency of the black carbon concentration on the mixing layer height for (a) the pre-monsoon (March – May 2013), (b) the monsoon (June – Sept. 2013), (c) the post-monsoon (Oct. – Nov. 2013) and (d) the winter (Dec. 2013 – Feb. 2014) season. The concentration refers to a (a) 92 m, (b) 82 m, (c) 85 m, (d) 74 m interval of the mixing layer height (see explanation in 3.3). The widths of the bars indicate the number of data points, as given in the legend in Fig. (d).


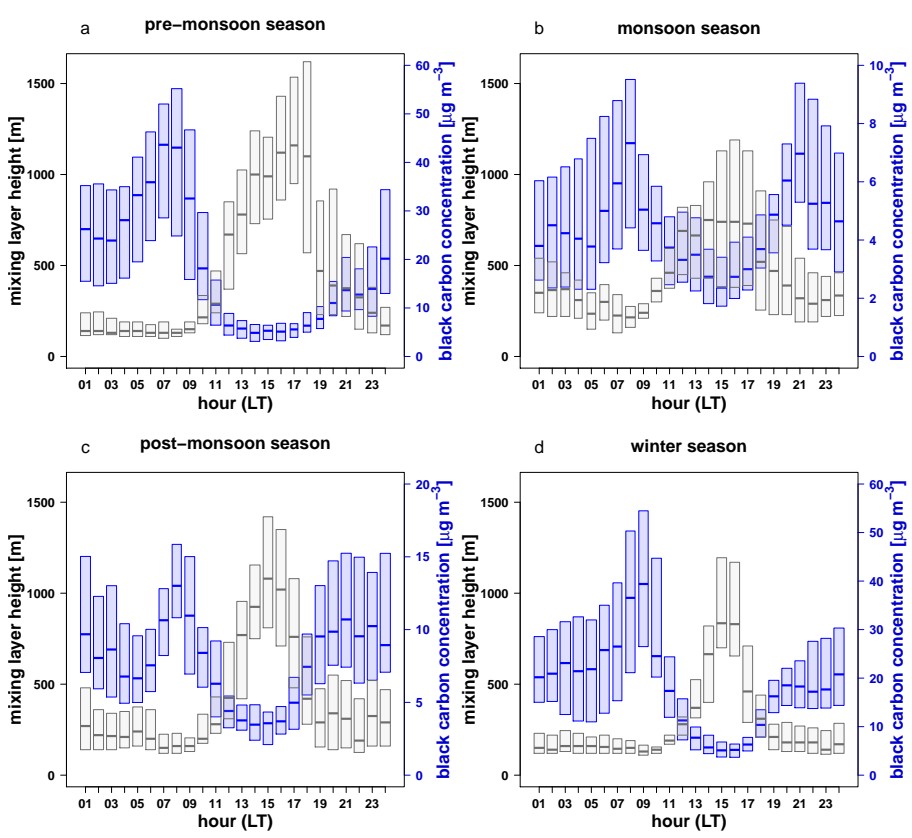

**Figure 9.** Diurnal cycle of the mixing layer height (grey) and the black carbon concentration (blue) as a Box-Plot (showing the median, the upper and lower quantile) for (a) the pre-monsoon (March – May 2013), (b) the monsoon (June – Sept. 2013), (c) the post-monsoon (Oct. – Nov. 2013) and (d) the winter (Dec. 2013 – Feb. 2014) season at the Bode station.

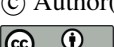



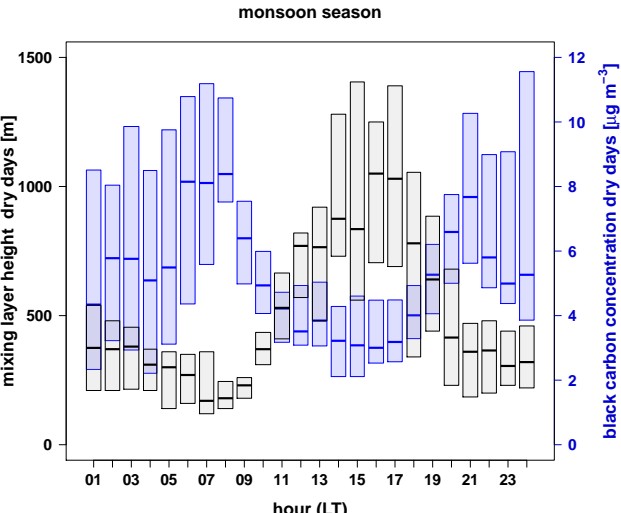

**Figure 10.** Diurnal cycle of the mixing layer height (grey) and the black carbon concentration (blue) for only dry days as a Box-Plot (showing the median, the upper and lower quantile) for the monsoon season (June – Sept. 2013) at the Bode station.





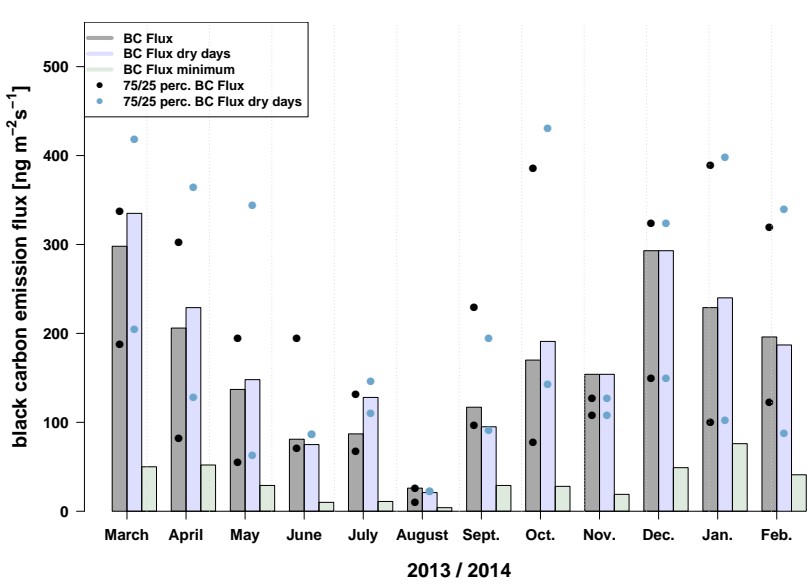

**Figure 11.** Seasonal cycle of the estimated black carbon emission flux based on the mean diurnal cycle per month using the data of all days ($F_{BC}$) (grey) and only dry days ($F_{BCdrydays}$) (blue) and with only using the estimated flux in the morning hours and for other hours zero flux is used ($F_{BCmin}$) (green) .