# Peer review of "Investigation of the mixing layer height derived from ceilometer measurements in the Kathmandu Valley and implications for local air quality"

_Atmospheric Chemistry and Physics, 2016_

## Referee Comment (RC1) · Anonymous Referee #1 · 27 Jan 2017

A year of mixing height estimates from a ceilometer are analysed and put into context to meteorological conditions observed at the surface. It is then used to enhance interpretation of measured black carbon concentration based on a simple model to calculate fluxes. Interesting results are presented and the impact of different seasons (e.g. Monsoon) and orography is discussed. However, the manuscript lacks some structure in terms some structure to put analysis and findings into a wider context. The introduction is focused on simple boundary layer knowledge, instead it should discuss mechanisms critical for the complex setup of the current study, such as the structure of the valley, predominant wind directions relative to moisture and pollution sources etc.

More literature references should be included to introduce the reader to the state of knowledge on boundary layer dynamics in valley locations and under Monsoon season conditions. This will allow the manuscript to clearly identify where new or contradicting findings are made.

Some more information on the uncertainty of the applied method for mixing height detection, its limitations and data availability should be included. Especially the point of excluding cloudy periods needs to be better explained in the context of the monsoon analysis.

Minor comments: P2, l15 – P3, l10: Very basic, it might be more interesting to reference other studies investigating the PBL to introduce current research in the field. Generally, much more literature references should be discussed in the introduction.

P2, l16: Might be the first study of this kind in Kathmandu Valley, but how about other locations with similar topography?

P4, l31: What type of weather station? Model and manufacturer of radiation sensor, temporal resolution? For all sensors provide sensor height and characteristics of immediate surroundings?

P5, l1: CL31 data also measured to hourly values? What is the original resolution in time and range?

P5, l10: State name of the algorithm.

P5, l25: Provide details on screening methods. How did you determine the cloud is 'within mixing height'? How do you define times with precipitation?

P5, l33: Does BLview have different versions? If so, which version of the software is used in this study?

P6, l5: How is the signal noise determined prior to averaging?

P6, l10: State explicitly: if aerosol load is low, BLview does not report any layer?

P6, l13: repetition with line 9

P6, l15: So BLview does not provide one estimate for mixing height but rather several to choose from? Explain the physical meaning.

P6, l20: At what time resolution is the tracking of layers performed?

P7, l2: Provide reference or reasoning for 'settings fitting best for mixing layer height assessment'

P7, l3: Not quite clear, it is stated that 'best settings' for mixing height detection are used, however, a resolution of 20 m is chosen even though a resolution of 10 m is required as stated in line 3. Check paragraph for consistency. Move whole paragraph to instrumentation section.

P7, l10: Data availability? How many hourly mixing height estimates are obtained within the study period? What are reasons for missing data? If all periods with boundary layers clouds are excluded from the analysis as stated in P5, l25, how representative are measurements during the monsoon season?

P7, l11: Comment on seasonal variations in sunrise and day-length.

P8, l7: What are the mixing height estimates during these 'very stable conditions'? Are those reasonable or is it possible that the detection algorithm has a limit for the lowest detectable height? What is the first range gate where BLview provides layer estimates?

P8, l15: What is the typical cloud base height of monsoon clouds over the region?

P8, l28: It is stated that reduction in solar energy is the dominant cloud effect, what is the physical explanation for the impact by rainfall?

P9, l9: Data availability of black carbon measurements? What data processing and quality control/instrument calibration is applied?

P9, l30: How are these temperatures determined? Include sensors in instrumentation

section.

P9, l35: More references to local dynamic characteristics in the valley should be provided in the introduction to 'set the scene' for this study, i.e. what are the results of previous studies, what are the open research questions?

P10, l20: Discuss local emission sources

P10, l25: Comment on the timing of rainfall during the day.

P11, l10: Comment on the implications of the site locations of the Bode site compared to the urban site. How is this related to timing of emissions? (see line 23: Make discussion on comparison to Puerto et al more consistent.)

P11, l30-10: Some of this should be moved to introduction.

P11, l26: Where is the pollution rose figure? Reference?

P12, l1: mark emission sources in map of Figure 1.

P13, l1: Comment on the validity of these assumptions? Discuss mechanisms in that influence advection and stratification in the valley. Reference relevant literature.

P13, l6: What are the hours used for the calculation? You state here no times with entrainment are used but then black carbon is used during morning hours – line 21. Is that consistent?
* * *

---

## Short Comment (SC1) · 23 Feb 2017

The authors analyzed the seasonal changes of mixing layer height in the Kathmandu Valley, calculated the black carbon emission fluxes, and presented some interesting results. Since our team also conducted some ceilometer observations, I feel very interested in your study, and there are some tiny questions that I want to discuss with the authors:

a. As for the estimation of mixing layer with a ceilometer, some extreme weather conditions should not be ignored, such as the windy and sand storm days. Under these

circumstances, the ceilometer retrivals may exist big errors [Tang et al., 2016]. Could the authors made some evaluations of the ceilometer measurements about this?

b. Since the black carbon is nonreactive, the black carbon column concentration in the mixing layer could represent the emission and transport contributions [Zhu et al., 2016]. If we ignore the transport effect, the diurnal variation in black carbon column concentration is consistent with the emission variations, thus can reveal the emission diurnal characteristics.

Tang, G., et al. (2016), Mixing layer height and its implications for air pollution over Beijing, China, Atmospheric Chemistry and Physics, 16(4), 2459-2475, doi:10.5194/acp-16-2459-2016.

Zhu, X., G. Tang, B. Hu, L. Wang, J. Xin, J. Zhang, Z. Liu, C. Münkel, and Y. Wang (2016), Regional pollution and its formation mechanism over North China Plain: A case study with ceilometer observations and model simulations, Journal of Geophysical Research: Atmospheres, 2016JD025730, doi:10.1002/2016JD025730.

---

## Referee Comment (RC2) · Anonymous Referee #2 · 13 Mar 2017

This MS describes a unique data set on the mixing height for a complete one year in the Kathmandu Valley and provides an essential information over this region. There are not many studies with such round the clock observations over the year period in this part of the world. However, I still see scope for a significant improvement in the MS.

Since there are very limited studies, it is better to provide some more information on the mixing height variations over this region. I strongly feel that it will be very good to show (Fig 3) monthly diurnal variation in-stead of seasonal. This will also provide a good

reference for a region with very complex topography. Additionally, average (sunrise, noon and sunset time) mixing height with 1 standard deviation can be provided for each month in a tabular form. Some of the specific and general comments are -

Abstract: Line 9-10: This is a common feature. It is better to add some quantitative information here. Like, height during night and day time, how does it changes with seasons?

Introduction: It includes very basic discussion on the boundary layer and it can be trimmed down.

Section 2.1: It is better to provide a brief description of BC instrument (Aethalometer) and if any data correction method is used.

Section 2.2: The Ceilometer is a commercial instrument and it has been used widely. Therefore, a brief mention of methodology adopted by others on mixing height determination and also its average reporting (from minutes to hours) can be provided.

Results:

Section 3.2: Fig 4: It would be useful to discuss briefly the differences in the diurnal patterns of solar radiation and mixing layer height. Peak of mixing layer height is about 3-4 hours later than the peak in solar radiation, why?

Section 3.3.1: Page 11, line 14-17: I presume that this correlation is determined using 24 hours average data. I feel that if this correlation is calculated for 2-3 time windows (morning, noon, evening etc), it will give better information.

General:

Page 3, line 11-15: These lines on ceilometer are not needed here and can be moved in to section 2.2.

Page 5, line 10-13: A reference for this comparison will provide a clear information to the readers. Briefly, outcome of the comparison can also be mentioned.

Figure 5 and 6: It is better to change the colour scheme. Yellow and green colours are not clearly visible.

---

## Author Comment (AC1) · 12 Apr 2017

Below we reply to the short comment by Dr. Tang on our ACPD manuscript "Investigation of the mixing layer height derived from ceilometer measurements in the Kathmandu Valley and implications for local air quality". We would like to thank Dr. Tang for his interest in our work and for the constructive comments helping us to improve the paper. We have listed Dr. Tang's comments below and answers are provided in blue. All page and line numbers refer to the "track changes" version of the revised manuscript provided as a supplement.

[Figure]

Short comment by G. Tang

The authors analyzed the seasonal changes of mixing layer height in the Kathmandu Valley, calculated the black carbon emission fluxes, and presented some interesting results. Since our team also conducted some ceilometer observations, I feel very interested in your study, and there are some tiny questions that I want to discuss with the authors:

a. As for the estimation of mixing layer with a ceilometer, some extreme weather conditions should not be ignored, such as the windy and sand storm days. Under these circumstances, the ceilometer retrivals may exist big errors [Tang et al., 2016]. Could the authors made some evaluations of the ceilometer measurements about this?

We checked our wind and precipitation data but could not find anything that stands out as unusual weather events. By excluding data from the ceilometer measurements if low clouds, precipitation or fog was observed (within the mixing layer) and by also excluding whole days with a precipitation sum of $> 0.5$ mm/day the impact of precipitation on the mixing layer height was considered and discussed in the study (especially for the monsoon season). Wind speed was found to be less than 6 m s$^{-1}$ most of the time and no data on dust were available. We therefore could not assess what difference it would make if severe weather days would be filtered out.

b. Since the black carbon is nonreactive, the black carbon column concentration in the mixing layer could represent the emission and transport contributions [Zhu et al., 2016]. If we ignore the transport effect, the diurnal variation in black carbon column concentration is consistent with the emission variations, thus can reveal the emission diurnal characteristics.

We agree with this point and added some more discussion to the manuscript on the assumption that horizontal and vertical transport can be neglected at night in order to determine top-down emissions (p. 16, l. 20-30). However, since transport can only be neglected during nighttime, when the boundary layer is very stable (little vertical mixing) and the horizontal wind speed is low (little horizontal mixing), but not during the daytime (when turbulent vertical entrainment and transport in and out of the valley passes occurs), the information is unfortunately insufficient to determine the diurnal variation in emissions. As shown in Berkes et al. (2016) the entrainment fluxes at the capping inversion can become substantial particularly if clouds are involved.

[Figure]

**Supplement:**

[revised manuscript text omitted]

---

## Author Comment (AC2) · 12 Apr 2017

Below we address the comments of reviewer #1 and questions raised during the open discussion of the paper "Investigation of the mixing layer height derived from ceilometer measurements in the Kathmandu Valley and implications for local air quality". We would like to thank the reviewer for the time and effort reviewing the paper. We feel it has improved thanks to the constructive comments. We have listed all reviewer comments below and our answers are provided in blue. Unless otherwise noted, all page and line numbers refer to the "track changes" version of the revised

manuscript provided as a supplement.

Anonymous Referee #1

However, the manuscript lacks some structure in terms some structure to put analysis and findings into a wider context. The introduction is focused on simple boundary layer knowledge, instead it should discuss mechanisms critical for the complex setup of the current study, such as the structure of the valley, predominant wind directions relative to moisture and pollution sources etc.

More literature references should be included to introduce the reader to the state of knowledge on boundary layer dynamics in valley locations and under Monsoon season conditions. This will allow the manuscript to clearly identify where new or contradicting findings are made.

We have revised the introduction as suggested and included more literature mainly on mixing layer height observation in similar terrains and the valley and on the circulation, predominant wind direction and characteristic local flows in the Kathmandu Valley. Furthermore, we shortened the part on the general description of the planetary boundary layer and the mixing layer height. Please, find the corresponding changes directly in the text (see track changes version).

Some more information on the uncertainty of the applied method for mixing height detection, its limitations and data availability should be included. Especially the point of excluding cloudy periods needs to be better explained in the context of the monsoon analysis.

An uncertainty estimation has not been applied in this study. To our knowledge there is only one publication available on confidence levels and error bars for the detection of mixing layer heights by Vaisala ceilometers (Münkel et al., 2011) which is based on a measurement campaign at Athens airport. It is not possible to derive general conclusions for other sites from this. As already discussed in section 2.2 results from ceilometer measurements are in good agreement with mixing layer height observations determined with other techniques such as sodar or radiosondes also in mountain areas (e.g. Ketterer et al., 2014).

References:

Münkel et al., 2011: Christoph Münkel ; Klaus Schäfer and Stefan Emeis, Adding confidence levels and error bars to mixing layer heights detected by ceilometer, Proc. SPIE 8177, Remote Sensing of Clouds and the Atmosphere XVI, 817708 (October 26, 2011); doi: 10.1117/12.898122; http://dx.doi.org/10.1117/12.898122.

Ketterer, C., Zieger, P., Bukowiecki, N., Collaud Coen, M., Maier, O., Ruffieux, D., and Weingartner, E.: Investigation of the Planetary Boundary Layer in the Swiss Alps Using Remote Sensing and In Situ Measurements, Boundary-Layer Meteorology, 151, 317-334, doi:10.1007/s10546-013-9897-8, 2014.

The part on the data availability together with the next question is answered in the replies to the "minor comments".

P7, l10: Data availability? How many hourly mixing height estimates are obtained

within the study period? What are reasons for missing data? If all periods with boundary layers clouds are excluded from the analysis as stated in P5, l25, how representative are measurements during the monsoon season?

Please see replies to "minor comments" below.

Minor comments:

P2, l15 - P3, l10: Very basic, it might be more interesting to reference other studies investigating the PBL to introduce current research in the field. Generally, much more literature references should be discussed in the introduction.

As mentioned above, we shortened the text on the planetary boundary layer and mixing layer in the introduction and included more literature references on local dynamics. Please, find the changes directly in the introduction (see track changes version).

P2, l16: Might be the first study of this kind in Kathmandu Valley, but how about other locations with similar topography?

We included a literature reference on mixing layer height detection from radio wind profiler at a station in the central Himalayas (India) (Singh et al., 2016). In this study measurements have been taken for five months (p.3, l. 26).

P4, l31: What type of weather station? Model and manufacturer of radiation sensor, temporal resolution? For all sensors provide sensor height and characteristics of

immediate surroundings?

As suggested, we included more detailed information on the weather station (p. 5, l. 26-31):

"The meteorological parameters were measured with an automatic weather station (Campbell Scientific, UK) with the CS215 sensor for humidity and temperature, the CS300 pyranometer for global radiation, and the RM Young 05103-5 wind vane to measure wind speed and direction. Global solar radiation observations were taken with an instrument measuring total sun and sky solar radiation. Its spectral range of 360 to 1120 nm encompasses most of the shortwave radiation that reaches the Earth's surface. The sensors were kept 2 m above the surface of the roof of a building (15 m above the ground). The meteorological data were recorded every minute."

P5, l1: CL31 data also measured to hourly values? What is the original resolution in time and range?

We added the following sentence to section 2.1 (p. 5, l. 21-24):

"The report interval of the CL31 is every 2 minutes with a vertical range resolution of 20 m. From that aerosol layer heights are calculated for every 10 minutes and mixing layer heights for every hour according to the description in section 2.2."

P5, l10: State name of the algorithm.

We included the name of the algorithm "Vaisala BL-VIEW" in parentheses in the text (p. 7, l. 18-20):

"In Münkel et al. (2011) measurements taken with eye-safe Vaisala ceilometers of the same kind as in the SusKat project were treated with the same algorithm (Vaisala BL-VIEW) to determine the mixing layer height as used in this study."

P5, l25: Provide details on screening methods. How did you determine the cloud is 'within mixing height'? How do you define times with precipitation?

We added a sentence to explain that in more detail (p. 8, l. 4-9):

"From the attenuated backscatter profiles, data points with observed clouds, fog or precipitation within the mixing layer height are excluded from the analysis. Therefore the degree of signal attenuation is so low that the attenuated backscatter profile can be used as a proxy for the aerosol density in the observed backscattering volume. The criterion to exclude data points with (i) clouds or fog, i.e., cloud bases or vertical visibility are below 160 m and with (ii) precipitation, i.e., cloud layers are below 4000 m and an attenuated backscatter signal of at least $2 \cdot 10^{-6}$ m$^{-1}$ sr$^{-1}$ in the height range 0 m - minimum (cloud base, 1000 m)."

P5, l33: Does BLview have different versions? If so, which version of the software is used in this study?

We included the version number in the corresponding sentence (p.8, l. 9-11):

"The algorithm used to determine the aerosol density and mixing layer height from ceilometer data collected in the Kathmandu Valley is the Vaisala BL-VIEW algorithm version 1.1, which has been slightly adapted according to the instrument settings used in this study."

P6, l5: How is the signal noise determined prior to averaging?

We included some sentences on p. 8, l. 22-26 to explain the determination of the signal noise:

"For noise detection the far range between 6200 m and 7200 m is investigated for every profile reported by the ceilometer. This far range profile contains 30 s average data. Two values are calculated for each profile in order to minimize the influence on high cloud signal: standard deviation of the profile values between 6200 m and 6700 m, and standard deviation of the profile values between 6700 m and 7200 m. The minimum of those values is the noise value used to determine the averaging time interval."

P6, l10: State explicitly: if aerosol load is low, BLview does not report any layer?

We added the following sentence on p. 8, l. 32-33:

"An aerosol layer is reported if a minimum threshold of $200 \cdot 10^{-9}$ m$^{-1}$ sr$^{-1}$ is exceeded in the attenuated backscatter signal."

P6, l13: repetition with line 9

We deleted this sentence.

P6, l15: So BLview does not provide one estimate for mixing height but rather several to choose from? Explain the physical meaning.

The lowest detected aerosol layer is regarded as the mixing layer height. We rephrased the sentence as follows (p. 9, l. 3-4):

> "For the evaluation in the frame of the SusKat project, the height of the lowest detected aerosol layer was regarded as the mixing layer height and the nocturnal stable boundary layer, respectively."

P6, l20: At what time resolution is the tracking of layers performed?

This is now mentioned on p. 5, l. 19-24:

> "In this study the measurements with a Vaisala ceilometer (CL31, Finland) (Münkel, 2007) are used to study the vertical structure of the atmosphere, especially the mixing layer height. To cover a whole annual cycle, data from March 2013 to February 2014 are used here. The report interval of the CL31 is every 2 minutes with a vertical range resolution of 20 m. From that aerosol layer heights are calculated for every 10 minutes and mixing layer heights for every hour according to the description in section 2.2."

P7, l2: Provide reference or reasoning for 'settings fitting best for mixing layer height assessment'

We added the following explanation (p. 9, l. 22-25):

"The settings fitting best for the mixing layer height assessment with the
CL31 are 16 s report interval and 10 m height resolution because the CL31
internal memory stores only up to 16 s accumulation time. These recom-
mended settings provide the best possible signal-to-noise ratio with the min-
imum required data flow."

P7, l3: Not quite clear, it is stated that 'best settings' for mixing height detection are
used, however, a resolution of 20 m is chosen even though a resolution of 10 m is
required as stated in line 3. Check paragraph for consistency. Move whole paragraph
to instrumentation section.

The text says that the instrument has *NOT* been operated with the settings fitting best
(p. 9, l. 25-26):

"During the time period investigated in this study, it had not been operated
with settings fitting best for mixing layer height assessment."

The 20 m instead of 10 m height resolution is an example for that.

P7, l10: Data availability? How many hourly mixing height estimates are obtained
within the study period? What are reasons for missing data? If all periods with
boundary layers clouds are excluded from the analysis as stated in P5, l25, how
representative are measurements during the monsoon season?

The data availability per season is given in Table 1. We included a sentence on that to
the corresponding paragraph (p. 6, l. 35 - p.7, l. 8):

"The availability of the mixing layer height data per hour varies between the

four seasons (70 to 93 %, Tab.1). Least data are available in the monsoon season which is likely due to a more frequent occurrence of clouds, fog and precipitation in this season, as the BL-VIEW algorithm excludes profiles with fog, precipitation or low clouds. We consider a data availability of 70 % as sufficient for our analysis."

P7, l11: Comment on seasonal variations in sunrise and day-length.

Here we collected some data on sunrise, sunset and day-length:

- Pre-monsoon season: sunrise between about 5 and 6:30 a.m., sunset between about 6 and 7 p.m., day-length between about 11.5 and 13.75 h

- Monsoon season: sunrise between about 5 and 6 a.m., sunset between about 5:50 and 7 p.m., day-length between about 12 and 14 h

- Post-monsoon season: sunrise between about 6 and 6:40 a.m., sunset between about 5:10 and 5:50 p.m., day-length between about 10.5 and 11.8 h

- Winter season: sunrise between about 6:30 and 7 a.m., sunset between about 5:10 and 6 p.m., day-length between about 10.5 and 11.5 h

We also included a sentence on that in the text (p. 10, l. 6-10):

"Sunrise varies between about 5 and 7 a.m. during the year with times between about 5 and 6:30 a.m. in the pre-monsoon season, 5 and 6 a.m. in the monsoon season, 5:10 and 5:50 a.m. in the post-monsoon season and 6:30 and 7 a.m. in the winter season. The day length during a year varies between about 10.5 and 14 hours, with 11.5 to 13.75 h in the pre-monsoon season, 12 to 14 h in the monsoon season, 10.5 to 11.8 h in the post-monsoon season and 10.5 to 11.5 h in the winter season."
P8, l7: What are the mixing height estimates during these 'very stable conditions'? Are those reasonable or is it possible that the detection algorithm has a limit for the lowest detectable height? What is the first range gate where BLview provides layer estimates?

The lowest gate where BL-VIEW provides mixing layer height estimates is 30 m. Because of the 20 m resolution setting used during SusKat, this configurable lower limit was set to 40 m.

P8, l15: What is the typical cloud base height of monsoon clouds over the region?

From the ceilometer data we could retrieve that during the monsoon season the monthly median cloud height was between 1360 m (June 2013) and 960 m (July and August 2013).

P8, l28: It is stated that reduction in solar energy is the dominant cloud effect, what is the physical explanation for the impact by rainfall?

We added some more explanation on p. 11, l. 31-33 and deleted "and rain" from the next sentence:

> "The sum of precipitation per day is used here as a proxy for the cloud amount to study the impact of days with a large fraction of cloud amount on the mixing layer height. Therefore the presence of clouds in the monsoon season explains a large part of the on average lower mixing layer height and higher variability compared with other seasons."

P9, l9: Data availability of black carbon measurements? What data processing and

quality control/instrument calibration is applied?

The data availability of black carbon measurements is given in Tab. 1. The analysis presented in the paper was based on the 1 min-average data collected at the Bode station. Post processing of the collected data includes flagging and averaging to 1-hr values. The 1-min data were flagged based on a few operational parameters of the instrument such as status, flow ratio, tape advance etc. in addition, the time series of black carbon was plotted against data from other co-located particle mass, number and trace gas concentration measurements to identify noise or influence of nearby sources. The contributions of nearby sources were identified by "short duration (1-3 minutes) spikes" in the time series, also indicated by other co-located instruments. The flags were applied to 1-min data, and the flagged data were averaged to 1-h values. The averaging only considers time period with at least 75 % (of 1-min) data points available as a valid 1-h values. For clarification, we added the following paragraph to the manuscript (p. 5, l. 33 - p. 6, l. 26):

"The black carbon concentrations were measured by an aethalometer model "AE33" (Magee Scientific, USA). This particular is a dual-spot measurement for loading correction (Drinovec et al., 2015). The AE33 deployed in Bode was fully factory calibrated before commencing the field campaign. In Bode, the AE33 was set to measure aerosol absorption at a time resolution of 1-min time average and the sample flow was set at local volumetric flow of 5 litres per minute. The tape advance was based on an attenuation of 120 and each tape advance was followed by clean air test. The clean air test was also performed whenever a new tape was inserted in the aethalometer. Operationally the AE33 is designed for a fairly automated and continuous measurement. Some of the operational issues during the field campaign include the change of the bypass filter cartridge, which was replaced whenever the clean air test failed or when the colour of the filter

inside the cartridge changed from white to grey or dark. Additionally, the optical chamber was also cleaned in response to the failure of the clean air test. The analysis presented in the paper was based on the 1-min average data collected at Bode. Post processing of the collected data includes flagging and averaging to 1-h values. The 1-min data was flagged based on operational parameters of the instrument such as status, flow ratio, tape advance etc. in addition, the time series of black carbon was plotted against data from other co-located particle mass, number and trace gas concentrations to identify noise or influence of nearby sources. The contributions of nearby sources were identified by "short duration (1-3 min) spikes" in the time series, also indicated by other co-located instruments. The flags were applied to 1-min data, and the flagged data were averaged to 1-h values. The averaging only considered time periods with at least 75 % (of 1-min) data points available as valid 1-h values."

P9, l30: How are these temperatures determined? Include sensors in instrumentation section.

Please, see comment above on the weather station.

P9, l35: More references to local dynamic characteristics in the valley should be provided in the introduction to 'set the scene' for this study, i.e. what are the results of previous studies, what are the open research questions?

The general description of the location of the Kathmandu Valley has been moved from section 2.1 to the introduction in order to introduce the studied region earlier in the text. Two more references to articles on local dynamics in the valley are provided in the introduction (Shrestha et al., 2015 and Regmi et al., 2003)

describing the main characteristic local flows in the Kathmandu Valley. Two more studies on the local dynamics are mentioned and described at the end of section 3.2. Please, find the corresponding changes directly in the text (see track changes version).

P10, l20: Discuss local emission sources

We added the following sentences to the text (p. 12, l. 24-27):

> "Important local emission sources of black carbon in the Kathmandu Valley are brick kilns which emit relatively constantly throughout day and night during the time period of December to April and are located in different parts of the valley (Fig. 1). Other important black carbon emission sources with a much more pronounced diurnal cycle are cooking, traffic (see location of main streets in Fig.1) and trash burning."

P10, l25: Comment on the timing of rainfall during the day.

The whole day is deleted from the time series if the sum of precipitation is above 0.5 mm day$^{-1}$ (p. 11, l. 23-25). For this reason, the exact timing of rainfall during the day has not been analysed in detail.

P11, l10: Comment on the implications of the site locations of the Bode site compared to the urban site. How is this related to timing of emissions? (see line 23: Make discussion on comparison to Puerto et al more consistent.)

We restructured section 3.3.1 to make the discussion more consistent (please see tack changes version).

[Figure]

P11, l30-10: Some of this should be moved to introduction.

Parts of this text are now moved to the introduction (please see track changes version).

P11, l26: Where is the pollution rose figure? Reference?

The pollution rose figures are not shown in the article, they were done only to check the relation between wind direction and black carbon concentration. We changed the corresponding sentence to (p. 15, l. 9-10):

> "Pollution roses (not shown) indicate that between about December and April most of the highest black carbon concentrations coincide with wind from the east and east south east."

P12, l1: mark emission sources in map of Figure 1.

We added the five major cities in the Kathmandu Valley (Kathmandu, Lalitpur, Bhaktapur, Kirtipur and Madhyapur-Thimi) to the revised version of figure 1, highlighted major roads, and marked clusters of brick kilns.

P13, l1: Comment on the validity of these assumptions? Discuss mechanisms in that influence advection and stratification in the valley. Reference relevant literature.

The assumption that the horizontal advection of air pollutants into and out of the valley at night is small is based on the observed low nocturnal wind speeds and on the

studies by Panday and Prinn (2009) and Panday et al. (2009), who found that the timing of the ventilation of air pollutants from the valley is largely determined by the strong westerly wind blowing through the valley from late morning until dusk. They conclude that the "bulk of ventilation takes place during the afternoon when strong westerly winds blow though the valley", which is consistent with our assumption that horizontal advection of air pollutants into and out of the valley is small at night.

The assumption that vertical mixing of pollutants between the mixing layer and the free troposphere is small at night is based on (i) the observations that the mixing layer height does not change significantly during this time of the day and (ii) the study by Regmi et al. (2003), who concluded that "during the nighttime, strong surface inversion and a deep cold-air lake would be formed under weak wind".

We added the main mechanisms and the references to the text (p. 16, l. 20-24 and l. 27-30):

> "The assumption of negligible horizontal transport at night is consistent with the observed low nocturnal wind speeds, and with the studies by Panday and Prinn (2009) and Panday et al. (2009), who found that the timing of the ventilation of air pollutants from the valley is largely determined by the strong westerly wind typically blowing through the valley from late morning until dusk. The authors of these studies conclude that the bulk of the ventilation of air pollutants usually takes place in the afternoon. Consequently, the horizontal advection of air pollutants into and out of the valley is typically small at night. [. . . ] The assumption that vertical mixing of pollutants between the mixing layer and the free troposphere is small at night is consistent with the findings by Regmi et al. (2003), who concluded that during night hours, the typically prevailing meteorology leads to a strong surface

inversion and formation of a deep cold-air lake in the valley under weak wind, which suppresses vertical mixing."

P13, l6: What are the hours used for the calculation? You state here no times with entrainment are used but then black carbon is used during morning hours - line 21. Is that consistent?

The times used in the estimation of the black carbon emission flux are given in Tab. 2. As described on p. 16 of the revised manuscript, the estimation of the black carbon flux is based on the conclusion that the main process driving the increase of the black carbon concentration during the night is not the variation in the mixing layer height (Fig. 9), nor mixing of even more polluted air from above or from outside the valley, but rather, as assumed here, the emissions. The variation of the mixing layer height during the times used for the calculation is only small (see Tab. 2 and Fig. 9).

To clarify that in the text we added a reference to table 2 and figure 9 to the corresponding sentence (p. 16, l. 24-27):

"And the mixing of pollutants between the mixing layer and the free atmosphere (entrainment) occurs mostly when the mixing layer height increases or decreases significantly, which is not the case during the hours used for this calculation (Tab. 2 and Fig. 9)."

---

## Author Comment (AC3) · 12 Apr 2017

Below we reply to the anonymous referee #2's comments and questions on our ACPD manuscript "Investigation of the mixing layer height derived from ceilometer measurements in the Kathmandu Valley and implications for local air quality". We would like to thank the reviewer for the constructive comments helping us to improve the paper. We have listed all reviewer comments below and answers are provided in blue. Unless otherwise noted, all page and line numbers refer to the "track changes" version of the revised manuscript provided as a supplement.

[Figure]

Anonymous Referee #2

This MS describes a unique data set on the mixing height for a complete one year in the Kathmandu Valley and provides an essential information over this region. There are not many studies with such round the clock observations over the year period in this part of the world. However, I still see scope for a significant improvement in the MS.

Since there are very limited studies, it is better to provide some more information on the mixing height variations over this region. I strongly feel that it will be very good to show (Fig 3) monthly diurnal variation in-stead of seasonal. This will also provide a good reference for a region with very complex topography. Additionally, average (sunrise, noon and sunset time) mixing height with 1 standard deviation can be provided for each month in a tabular form.

Following the suggestion of the reviewer, we now show the diurnal cycle of MLH in the revised version of figure 3 (see below) for each month separately. Additionally, we included the daylight hours for the 15th of each month as yellow shading also showing the sunrise and sunset times. As median MLH and variability at sunrise, sunset and noon are now shown in figure 3, we think an additional table as proposed by the reviewer is not needed.

Some of the specific and general comments are - Abstract: Line 9-10: This is a common feature. It is better to add some quantitative information here. Like, height during night and day time, how does it changes with seasons?

We added the following sentences to the abstract (p. 1, l. 10-13):

"The monthly minimum median MLH values typically range between 150 and 200 m during night and early morning hours, the monthly maximum median values between 625 m in July and 1460 m in March. Seasonal differences are not only found in the absolute mixing layer heights, but also in the duration of the typical daytime maximum ranging between 2-3 hours in January and 6-7 hours in May."

Additionally, we added numbers to the following sentence in the abstract (p. 1, l. 13-16):

"During the monsoon season a diurnal cycle has been observed with the smallest amplitude (typically between 400 and 500 m), with the lowest daytime mixing height of all seasons (maximum monthly median values typically between 600 - 800 m), and also the highest nighttime and early morning mixing height of all seasons (minimum monthly median values typically between 200 and 220 m)."

Introduction: It includes very basic discussion on the boundary layer and it can be trimmed down.

Following the suggestion of the referee, we shortened the part on the general description of the planetary boundary layer and the mixing layer height in the introduction (see track changes version).

Section 2.1: It is better to provide a brief description of BC instrument (Aethalometer) and if any data correction method is used.

We added the following description to the manuscript (p. 5, l. 34 - p. 6, l. 7):

"The aethalometer is among the earliest BC measurement methods and has been applied since the early 1980's (e.g. Hansen et al., 1982, 1984). This measurement is a filter-based method where air is drawn through a sampling filter and an increase in attenuation is detected with increasing aerosol loading on the filter. Its advantage is that it provides an absorption derived real time estimate of black carbon. However, the instrument measures absorption coefficient by all components of the aerosol besides black carbon over the broad region of visible spectrum. It requires therefore knowledge on the mass specific absorption cross section (MAC) of the BC-containing aerosol, which introduces some uncertainty to the measured values. The derived quantity is commonly referred to as "equivalent black carbon" (EBC), which is the case if the MAC is exactly known (see e.g Petzold et al., 2013)."

New references:

Hansen, A. D. A., Rosen, H., and Novakov, T.: Real-time measurement of the aerosol absoprtion-coefficient of aerosol particles, Appl. Opt., 21, 3060-3062, 1982.

Hansen, A. D. A., Rosen, H., and Novakov, T.: The aethalometer - an instrument for the real-time measurement of optical absorption by aerosol particles, Sci. Total Environ., 36, 191-196, 1984.

Petzold, A., Ogren, J. A., Fiebig, M., Laj, P., Li, S.-M., Baltensperger, U., Holzer-Popp, T., Kinne, S., Pappalardo, G., Sugimoto, N., Wehrli, C., Wiedensohler, A., and Zhang, X.-Y.: Recommendations for reporting "black carbon" measurements, Atmos. Chem. Phys., 13, 8365-8379, doi:10.5194/acp-13-8365-2013, 2013.

Section 2.2: The Ceilometer is a commercial instrument and it has been used widely. Therefore, a brief mention of methodology adopted by others on mixing height determination and also its average reporting (from minutes to hours) can be provided.

We added the following description to section 2.2 (p. 7, l. 13-16):

"Haeffelin et al. (2012) discuss the most common methods for mixing height determination with ceilometers; these include gradient methods investigating first or second derivative of the backscatter profile reported by the instrument, backscatter variance, wavelet and backscatter profile covariance, and fitting of ideal backscatter profiles. All methods involve temporal averaging ranging from 2 to 60 minutes, depending on the atmospheric conditions and the performance of the instrument."

New reference:

Haeffelin, M., Angelini, F., Morille, Y., Martucci, G., Fry, S., Gobbi, G. P., Lolli, S., O'Dowd, C. D., Sauvage, L., Xueref-Rémy, I., Wastine, B., and Feist, D. G.: Evaluation of Mixing-Height Retrievals from Automatic Profiling Lidars and Ceilometers in View of Future Integrated Networks in Europe, Boundary-Layer Meteorology, 143, 49-75, doi: 10.1007/s10546-011-9643-z, 2012.

Results:

Section 3.2: Fig 4: It would be useful to discuss briefly the differences in the diurnal patterns of solar radiation and mixing layer height. Peak of mixing layer height is about 3-4 hours later than the peak in solar radiation, why?

The explanation of this can be found in various textbooks. Here a condensed description of the processes is taken from chapter 9 of "Atmospheric Science" by Wallace and Hobbs:

The reason for difference in the diurnal timing can be understood considering the growth process of the mixed layer. Solar radiation heats the Earth's surface. Due to the heating capacity of the ground, the temperature increase in the surface ground layer lags behind the increase in solar radiation during the day, and continues on after noon until it peaks in the afternoon, when the incoming solar radiation has decreased enough that the surface ground layer begins to cool again. After sunrise, the warmed surface ground layer in turn warms the overlying air in the so-called "surface layer" (which is roughly the lowest 5% of the mixed layer depth). This typically results in a superadiabatic vertical temperature gradient, i.e., an unstable vertical layering. Small turbulent motions (e.g., overturning due to wind shear) initiate mixing, and once set in vertical motion, the heated parcels of air from the surface layer rise dry convectively, cooling adiabatically, until they reach their level of neutral buoyancy (determined by the difference in the ambient lapse rate and the dry adiabatic lapse rate), which over time becomes the capping layer of the vertically growing boundary layer. Due to the kinetic energy obtained while vertically accelerating, the heated parcels from the surface overshoot their level of neutral buoyancy, and simultaneously force air from the free troposphere down into the mixing layer. The overshooting parcels in a generally laminar free troposphere can remain largely intact and sink back down into the mixing layer. However, the air masses from the free troposphere that are forced down into the mixing layer are "immediately torn and mixed into the mixed layer by the strong turbulence there. . . [and] become one with the mixed layer and never return to the free atmosphere." This entrainment, which defines the entrainment zone, is the process by which the mixing layer grows. "It can be thought of as a mixed layer that gradually eats its way upward into the overlying air." Because this process takes time for the growth to occur, and also because of the lag described above between the peak in the
surface ground temperature and the incoming solar radiation, the peak in the mixing layer height is in the late afternoon, as contrasted with the noon peak in incoming solar radiation (Figure 4).

In Section 3.2 of the submitted version of the paper (p. 8, l. 3-4), this was all indicated with the single sentence:

> "This is due to a delayed response in the production of thermal turbulence to the warming of the ground by the incoming solar radiation."

Since this is evidently too brief, but the full description above would be a bit disproportionately long to include in the paper, we have replaced the former brief sentence with the following (p. 11, l. 1-8):

> "This is due to the growth process of the mixing layer, which is driven by the heating of the ground by incoming solar radiation. This heating causes thermals to rise from the surface layer, causing mixing when they overshoot into the more stable free troposphere at the entrainment layer, which in turn results in growth of the mixing layer by gradual assimilation of overlying free tropospheric air which is forced down into the mixing layer throughout the day. Because it takes time for the mixed layer to grow by this process, and also because the increase in the surface ground temperature lags behind the increase in incoming solar radiation during the morning, the peak in the mixing layer height is in the late afternoon, as contrasted with the noon peak in incoming solar radiation."

Section 3.3.1: Page 11, line 14-17: I presume that this correlation is determined using 24 hours average data. I feel that if this correlation is calculated for 2-3 time windows

(morning, noon, evening etc), it will give better information.

The temporal correlations given in section 3.3.1 have been calculated from the MLH and BC time series with a time resolution of 1 hour. The temporal correlations shown here reflect to a large degree the diurnal cycle. Further splitting the time series into different times of the day in addition to splitting the time series into different seasons would result in a further reduction of the correlation. Here, we show the differences between the seasons, we would therefore prefer to not split the time series into different times of the day. However, we clarified the calculation of the temporal correlations given in section 3.3.1 by rewriting the corresponding sentence and adding the following explanation (p. 12, l. 18-22):

"The correlation coefficients calculated for the time series of mixing layer height and black carbon concentration with a time resolution of 1 hour depend strongly on the season. While the pearson coefficient for the pre-monsoon season is -0.54, it is only -0.19 for the monsoon season (-0.40 for the post-monsoon and -0.46 for the winter season). This shows that a part of the variation in time of the black carbon concentration can be explained by atmospheric dynamics and that its magnitude depends on the season."

General:

Page 3, line 11-15: These lines on ceilometer are not needed here and can be moved in to section 2.2.

We agree and shortened these lines as follows (leaving a reference to section 2.2) (p. 4, l.12-16):

"A ceilometer was deployed to measure vertical profiles of the aerosol attenuated backscattering during the SusKat-ABC campaign at Bode, the supersite of the SusKat-ABC campaign, located in a semi-urban setting in the Kathmandu Valley (more details are presented in section 2.2)."

Page 5, line 10-13: A reference for this comparison will provide a clear information to the readers. Briefly, outcome of the comparison can also be mentioned.

Following the suggestion of the reviewer, we added more details on the study Münkel et al. (2011) replacing lines 10-13 on page 5 of the original manuscript (p. 7, l. 21-26):

"In particular, Münkel et al. (2011) compare the mixing layer heights obtained during the Tall Wind measurement campaign in Hamburg, Germany, and routine measurements carried out in Vantaa, Finland, with mixing layer heights derived from potential temperature and relative humidity profiles reported by radio soundings. The example cases presented in the study show a good agreement with deviations not exceeding 10%."

Figure 5 and 6: It is better to change the colour scheme. Yellow and green colours are not clearly visible.

We adjusted the colors of figures 5 and 6 in order to improve legibility. In particular the contrast between overlapping and non-overlapping parts of the boxes has been increased.